# FLY-CL: A FLY-INSPIRED FRAMEWORK FOR ENHANCING EFFICIENT DECORRELATION AND REDUCED TRAINING TIME IN PRE-TRAINED MODEL-BASED CONTINUAL REPRESENTATION LEARNING

**Heming Zou**[1]* **Yunliang Zang**[2]* **Wutong Xu**[1] **Xiangyang Ji**[1]†

[1]Department of Automation, Tsinghua University
[2]Academy of Medical Engineering and Translational Medicine, Tianjin University
`{zouhm24, xwt22}@mails.tsinghua.edu.cn`
`yunliangzang@tju.edu.cn, xyji@tsinghua.edu.cn`

## ABSTRACT

Using a nearly-frozen pretrained model, the continual representation learning paradigm reframes parameter updates as a similarity-matching problem to mitigate catastrophic forgetting. However, directly leveraging pretrained features for downstream tasks often suffers from multicollinearity in the similarity-matching stage, and more advanced methods can be computationally prohibitive for real-time, low-latency applications. Inspired by the fly olfactory circuit, we propose Fly-CL, a bio-inspired framework compatible with a wide range of pretrained backbones. Fly-CL substantially reduces training time while achieving performance comparable to or exceeding that of current state-of-the-art methods. We theoretically show how Fly-CL progressively resolves multicollinearity, enabling more effective similarity matching with low time complexity. Extensive simulation experiments across diverse network architectures and data regimes validate Fly-CL's effectiveness in addressing this challenge through a biologically inspired design. Code is available at https://github.com/gfyddha/Fly-CL.

## 1 INTRODUCTION

Artificial neural networks have exhibited remarkable capabilities across various domains in recent years. Nevertheless, real-world applications often require continuous model adaptation to handle progressively emerging unseen scenarios, making updates based on sequential incoming data essential. This need has led to the development of Continual Learning (CL). Earlier research primarily focused on training models from scratch (Aljundi et al., 2018; Kirkpatrick et al., 2017; Li & Hoiem, 2017; Zenke et al., 2017). Pretrained models have recently become prominent in CL, owing to their robust generalization in downstream tasks for downstream tasks (Wang et al., 2022a;b; Zou et al., 2025b).

Popular CL methods utilizing pre-trained models can generally be classified into three categories: (1) prompt/adapter-based approaches (Jung et al., 2023; Smith et al., 2023; Tang et al., 2023; Wang et al., 2022a;b; 2025a; Liang & Li, 2024; Yu et al., 2024; Xiao et al., 2025; Wang et al., 2024b; 2025c), (2) mixture-based approaches (Chen et al., 2023; Gao et al., 2023; Wang et al., 2023a;b; Zhou et al., 2023b; Che et al., 2025), and (3) representation-based approaches (McDonnell et al., 2023; Sun et al., 2025; Zhou et al., 2023a; 2024; Zhuang et al., 2024). All three paradigms operate without exemplars and significantly outperform traditional training-from-scratch methods. Despite their strengths, each approach has limitations. Prompt/adapter-based methods are constrained to transformer architectures, and updating prompts/adapters inherently risks propagating forgetting within the prompt/adapter space. Mixture-based approaches require storing previous models, resulting in significant storage overhead and increased computational complexity during model fusion. Compared with parameter-learning approaches, **representation-based methods perform better by reframing**

---

*Equal contribution
†Corresponding author

**learning as similarity matching and avoiding dependence on a specific backbone**. They form class prototypes (CPs) by averaging features extracted by a frozen pretrained network, with each prototype acting as the centroid of its class. However, **insufficient separation between features can lead to ambiguous decision boundaries when distinguishing between prototypes, complicating the similarity matching process**. This challenge is formally recognized as the multicollinearity problem. Previous studies (McDonnell et al., 2023) have attempted to address this issue, but they still incur substantial computational costs, hindering real-world deployment.

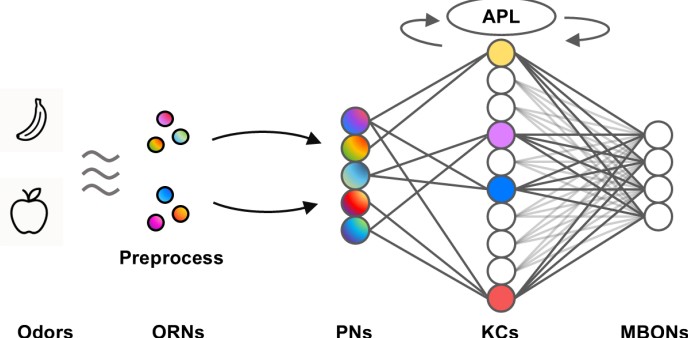

Figure 1: **Schematic of the Fly Olfactory Circuit.** Odors are first detected and pre-processed by olfactory receptor neurons (ORNs) in the antennal lobe, where feature extraction and normalization take place, before being transmitted to projection neurons (PNs). Expansion coding occurs as signals move from PNs to Kenyon cells (KCs), with an expansion ratio of approximately 40. Each KC connects to a fixed number of PNs (about 6). Lateral inhibition, mediated by an anterior paired lateral (APL) neuron, suppresses most weakly activated KCs, exemplifying a winner-take-all strategy. Finally, signals from KCs to mushroom body output neurons (MBONs) involve a dense down-projection that reduces dimensionality to select specific actions.

In biological systems, pattern separation is a well-recognized circuit motif across cerebellum-like and related networks, including the cerebellum, hippocampus, and fly olfactory system (Lin et al., 2014; Papadopoulou et al., 2011; Stevens, 2015; Zang & De Schutter, 2023). These neural circuits efficiently decorrelate overlapping sensory inputs. Figure 1 illustrates the information processing mechanism in the fly olfactory circuit. The process unfolds in several steps: in response to an odor stimulus, olfactory neurons extract and pre-process odor information represented as a 50-dimensional vector (PNs). These pre-extracted features are then randomly projected into expanded dimensions (KCs), selectively activating a small subset of KCs that receive the strongest excitation while zeroing out others. The high-dimensional features in KCs subsequently converge to low-dimensional MBONs for classification. Numerous studies have highlighted the role of the PN→KC transformation in producing decorrelated representations, and many strategies have been proposed to model this process. By contrast, the downstream KC→MBON transformation has received far less attention, and a clear theory of its contribution to decorrelation has yet to be established. To address this gap, **we theoretically show that the KC→MBON pathway also supports decorrelation under the Hebbian learning rule by demonstrating its consistency with ridge classification (See Section 6)**.

Inspired by the fly olfactory circuit, we propose Fly-CL, an efficient framework for progressive decorrelation in representation-based learning with pre-trained models. The pipeline starts with feature extraction and normalization, followed by high-dimensional random sparse projection and a top-$k$ operation for collective decorrelation, mimicking the PN→KC process. We then implement a prototype similarity matching mechanism mimicking the KC→MBON structure during inference, and an efficient streaming ridge classification method to decorrelate parameter weights during training. Empirical results show substantially reduced time consumption versus baseline methods.

Our main contributions are as follows:

1. We propose an efficient and biologically plausible decorrelation framework that significantly reduces computational costs compared to current SOTA methods while achieving comparable or improved performance in CL.

2. Our method's effectiveness and robustness in decorrelation are validated by extensive experiments under various data setups and model architectures, supported by theoretical and empirical analyses.

3. The alignment of our framework with the fly olfactory circuit suggests that biological structures can inspire effective and efficient solutions to AI problems.

## 2  RELATED WORK

**Representation-based methods in CL using Pre-trained Models:**   Representation-based methods (McDonnell et al., 2023; Sun et al., 2025; Zhou et al., 2023a; 2024) demonstrate superior performance over parameter-learning approaches by leveraging features extracted from a frozen pre-trained model to compute similarities with class prototypes. They are also more practical for resource-constrained deployments (e.g., edge computing), since they avoid updating the entire model. For instance, in a smart camera system, such methods enable the efficient addition of new object recognition capabilities without retraining the entire model. While showing promising progress, their computational overhead remains substantial, which may limit their applicability in scenarios requiring real-time responses.

**Fly Olfactory Circuit:**   Information processing in cerebellum-like circuits, including the fly olfactory circuit, involves several stages (Lin et al., 2014; Papadopoulou et al., 2011; Stevens, 2015; Zang & De Schutter, 2023). Theoretical neuroscience studies suggest that most stages exhibit progressive feature separation effects (Hige et al., 2015). Algorithms inspired by the fly olfactory circuit have been applied across various AI domains, including parameter-efficient fine-tuning (Zou et al., 2025c), continual learning (Zou et al., 2025b), locality-sensitive hashing (Dasgupta et al., 2017; Sharma & Navlakha, 2018), word embedding (Liang et al., 2021), and federated learning (Ram & Sinha, 2022).

## 3  BACKGROUND

### 3.1  PROBLEM STATEMENT

In this paper, we focus on CL within the context of image classification tasks. We denote sequentially arriving tasks as $\mathcal{D} = \{\mathcal{D}_1, \ldots, \mathcal{D}_T\}$, where each task $\mathcal{D}_t = \{(\boldsymbol{x}_t^i, y_t^i)\}_{i=1}^{n_t}$ consists of $n_t$ samples. Each sample $\boldsymbol{x}_t^i$ within a task is drawn from the input space $\mathcal{X}_t$, and its corresponding label $y_t^i$ belongs to the label space $\mathcal{Y}_t$ with $c_t$ classes. The training process involves sequential learning from $\mathcal{D}_1$ to $\mathcal{D}_T$, followed by class prediction on an unseen test set spanning the full label space.

To demonstrate the effectiveness and efficiency of our framework, we adopt Class Incremental Learning (CIL), a widely used experimental setup. Unlike traditional Task Incremental Learning, CIL does not provide access to the task ID during the testing process, making it more challenging. In CIL, the model learns mutually exclusive classes within each task, ensuring that the intersection of label spaces satisfies $\mathcal{Y}_i \cap \mathcal{Y}_j = \emptyset$. We denote that there are $c_t$ classes for the first $t$ tasks.

### 3.2  REPRESENTATION-BASED PARADIGM IN CL

We study on the basis of the recently popular representation-based paradigm using pre-trained models (McDonnell et al., 2023; Sun et al., 2025; Zhou et al., 2023a; 2024), which demonstrates superior performance for CL. Given an input image $\boldsymbol{x}_t^i$, it is first compressed into a $d$-dimensional feature $\boldsymbol{v}_t^i = f_\theta(\boldsymbol{x}_t^i) \in \mathbb{R}^d$ using a pre-trained encoder $f_\theta$. For each class $i$ in task $t$, we compute its prototype by averaging features over all training samples belonging to this class:

$$\boldsymbol{\mu}_t^i = \frac{1}{N_t^i} \sum_{j=1}^{|\mathcal{D}_t|} \mathbb{I}(y_t^j = i) f_\theta(\boldsymbol{x}_t^j) \in \mathbb{R}^d, \tag{1}$$

where $N_t^i = \sum_{j=1}^{|\mathcal{D}_t|} \mathbb{I}(y_j = i)$ denotes the cardinality of class $i$'s training set, and $\mathbb{I}(\cdot)$ is the indicator function. During inference, for a test sample with feature vector $\boldsymbol{v}$, the predicted class $\hat{y}$ is determined by finding the maximum cosine similarity between $\boldsymbol{v}$ and all learned class prototypes:

$$\hat{y} = \arg\max_{t,i} \frac{\boldsymbol{v}^\top \boldsymbol{\mu}_t^i}{\|\boldsymbol{v}\| \cdot \|\boldsymbol{\mu}_t^i\|}. \tag{2}$$

However, significant inter-prototype correlations ($\mathbb{E}[\boldsymbol{\mu}_{t_i}^{c_t,i}{}^\top \boldsymbol{\mu}_{t_j}^{c_t,j}] \gg 0$, see Figure 3(a)), can severely compromise the discriminative power of the similarity measurement (Belsley et al., 2005). This phenomenon arises because high correlations reduce the effective angular separation between classes

in the embedding space, leading to ambiguous decision boundaries. Specifically, when prototypes cluster near a dominant direction in $\mathbb{R}^d$, the cosine similarity metric becomes less sensitive to subtle but critical inter-class distinctions, thereby degrading classification performance.

# 4 FLY-CL

In this section, we detail the design motivation and functionality of each component of our Fly-CL framework. A schematic of the overall framework is provided in Figure 2, along with the pseudocode for the training and inference pipeline in Appendix A. The decorrelation effect of each component is visualized in Figure 3 using Pearson correlation coefficients of different class prototypes.

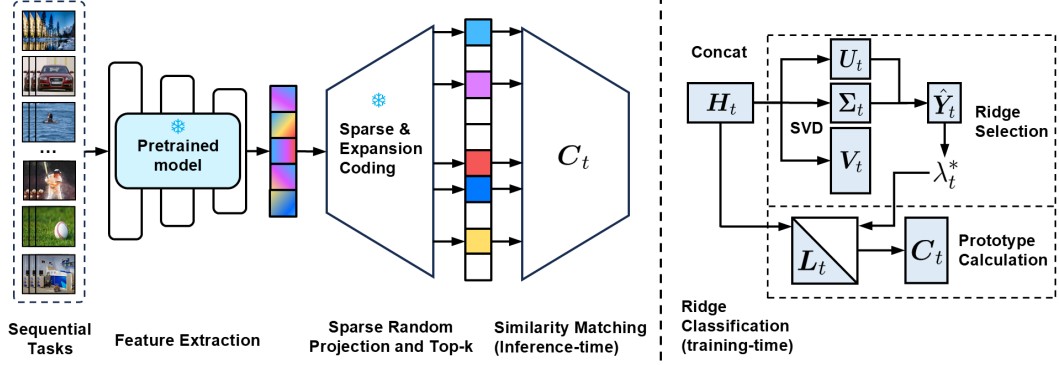

Figure 2: **Schematic of the Fly-CL Framework. Left:** Our framework extracts image embeddings using a frozen pre-trained model, projects them into a higher-dimensional space via a fixed sparse random projection, and filters them through a top-$k$ operation (PNs → KCs). Then we utilize a learned down-projection for similarity matching during inference time (KCs → MBONs). **Right:** During the training phase, the parameter $C_t$ is learned via a streaming ridge classification scheme.

## 4.1 SPARSE RANDOM PROJECTION AND TOP-K OPERATION

Building upon the representation-based paradigm, it is necessary to decouple different class prototypes. Inspired by the decorrelation mechanism of the fly olfactory circuit, we emulate the sparse expansion projection from PNs to KCs, followed by winner-take-all inhibition mediated by APL neurons. Given a feature embedding $v \in \mathbb{R}^d$ extracted from the pre-trained encoder, we formulate the transformation $Z(v) : \mathbb{R}^d \to \mathbb{R}^m$ as:

$$h' = Z(v) = \text{top-}k(h) = \text{top-}k\left(Wv\right), \tag{3}$$

where the fixed projection matrix $W \in \mathbb{R}^{m \times d}$ (with $m \gg d$) implements weight sparsity: each row contains exactly $p$ $(p < d)$ non-zero entries independently sampled from $\mathcal{N}(0, 1)$. The top-$k$ operator implements activation sparsity by preserving only the $k$ largest components $(k < m)$ while zeroing out others, formally defined as:

$$[h']_i = \begin{cases} [h]_i & \text{if the magnitude of } [h]_i \text{ is among the top-}k \text{ values of } h, \\ 0 & \text{otherwise.} \end{cases} \tag{4}$$

This two-stage process achieves effective decorrelation through the following properties, and its empirical effect is visualized in the transformation from Figure 3(a) to (b).

1. **High-Dimensional Embedding Enhances Linear Separability**: Random projection of low-dimensional features into an extremely high-dimensional space can improve the linear separability of the feature representations (Litwin-Kumar et al., 2017).

2. **Powerful Inhibition Suppresses Noisy Components**: The top-$k$ operation imposes sparsity by suppressing noisy dimensions that may interfere with discrimination through dimensional competition, while enhancing separation by keeping the most discriminative dimensions (Metwally et al., 2006).

Considering computational efficiency, $W$'s sparse pattern reduces the time complexity for random projection from $\mathcal{O}(mn_t d)$ to $\mathcal{O}(mn_t p)$ while preserving the core representational capacity compared to dense projection. Similarly, the top-$k$ operation reduces similarity matching complexity from $\mathcal{O}(mn_t c_t)$ to $\mathcal{O}(kn_t c_t)$, while simultaneously improving performance.

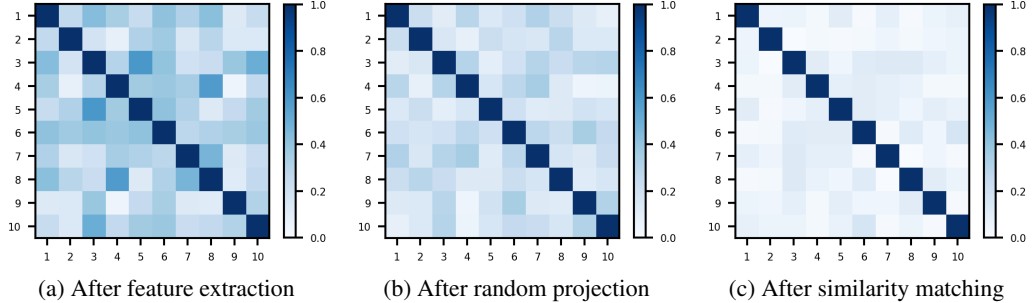

(a) After feature extraction     (b) After random projection     (c) After similarity matching

Figure 3: **Pearson Correlation Coefficients of Prototypes at Different Decorrelation Stages in Fly-CL.** Heatmaps display Pearson correlation coefficients for 10 randomly selected class prototypes at each stage of our pipeline (consistent across visualizations).

We further propose two theorems to demonstrate that strong sparsity does not significantly degrade performance. According to Theorem 4.1, as long as $p$ and $d$ are not extremely small, the matrix $\boldsymbol{W}$ retains full column rank with probability $1 - o(1)$, which is a common approach to demonstrate that a sparse random projection does not result in severe information loss. Furthermore, in Theorem 4.2, by proving that the performance degradation between this sparsification operation and the original vector is bounded, we show that if $k$ is not extremely small, it can preserve most of its performance. For a complete proof, please refer to Appendix B.

**Theorem 4.1.** *Given the matrix $\boldsymbol{W} \in \mathbb{R}^{m \times d}$, where $m > d$, with each row having exactly $p$ non-zero entries, which are randomly sampled from $\mathcal{N}(0, 1)$. Let $\mathcal{W} \in \mathbb{R}^{d \times d}$ be any square submatrix of $\boldsymbol{W}$. Then, for any $\epsilon > 0$, it holds that*

$$\mathbb{P}\left(|\det(\mathcal{W})| \geq \left(\frac{p}{d}\right)^{d/2} \sqrt{d!} \exp(-d^{1/2+\epsilon})\right) = 1 - o(1).$$

*Thus, for sufficiently large $p$ and $d$, any submatrix $\mathcal{W}$ is invertible with probability at least $1 - o(1)$.*

**Theorem 4.2.** *For top-$k$ sparsification in the expanded dimension $m$, the performance degradation is bounded by:*

$$\mathbb{E}\left[|L(\boldsymbol{h}, y) - L(\boldsymbol{h}', y)|\right] \leq M \cdot \sqrt{\frac{C}{k} \cdot \mathbb{E}[\|\boldsymbol{h}\|_2^2]},$$

*where $L(\cdot)$ is a performance loss function for downstream tasks and $C$, $M$ are constants. To ensure negligible performance degradation, we require:*

$$\sqrt{\frac{C}{k} \cdot \mathbb{E}[\|\boldsymbol{h}\|_2^2]} \leq \mathcal{O}\left(\frac{1}{\sqrt{m^\alpha}}\right),$$

*i.e., when $k = \Omega(m^\alpha)$ $(0 < \alpha < 1)$, the error bound decays polynomially with increasing dimension.*

### 4.2 STREAMING RIDGE CLASSIFICATION

Previous studies on decorrelation in the fly olfactory circuit have primarily focused on PNs→KCs transformation, where sparse and decorrelated representations have been experimentally observed. In contrast, the downstream KCs→MBONs transformation has received little attention, and the physiological evidence remains inconclusive. This motivates us to investigate whether the KC→MBON pathway also facilitates decorrelation. To simulate the training dynamics of the KC→MBON projections, we employ a biologically plausible Hebbian learning rule. As shown in Section 6, this rule is mathematically equivalent to ridge classification under the CL setting. Consequently, we implement an online formulation with adaptive regularization, which naturally achieves decorrelation while ensuring computational efficiency and compatibility with sequential data. Ridge classification (Hoerl & Kennard, 1970) mitigates feature collinearity through $\ell_2$-regularization, trading increased bias for reduced variance by shrinking correlated feature weights, thereby stabilizing prototype estimation in non-i.i.d. sequential learning scenarios. Let $\boldsymbol{H}_t \in \mathbb{R}^{n_t \times m}$ denote the concatenation of high-dimensional features $\boldsymbol{h}'$ for $n_t$ samples in task $t$, and $\boldsymbol{Y}_t \in \{0, 1\}^{n_t \times c_t}$ represent the corresponding one-hot label matrix for $c_t$ classes. We maintain two streaming statistics: a Gram matrix $\boldsymbol{G} \in \mathbb{R}^{m \times m}$,

capturing self-correlation, and a matrix $\boldsymbol{S} \in \mathbb{R}^{m \times c_t}$, accumulating cross-dimensional weights for each class prototype. During each task iteration $t$, these are updated as follows:

$$\boldsymbol{G}_t \leftarrow \boldsymbol{G}_{t-1} + \boldsymbol{H}_t^\top \boldsymbol{H}_t, \qquad \boldsymbol{S}_t \leftarrow \boldsymbol{S}_{t-1} + \boldsymbol{H}_t^\top \boldsymbol{Y}_t. \tag{5}$$

The classifier matrix $\boldsymbol{C} \in \mathbb{R}^{m \times c_t}$ is updated via regularized least squares accordingly:

$$\boldsymbol{C}_t = (\boldsymbol{G}_t + \lambda \boldsymbol{I}_m)^{-1} \boldsymbol{S}_t. \tag{6}$$

Prediction for preprocessed new samples $\boldsymbol{h}' \in \mathbb{R}^m$ follows:

$$\hat{y} = \arg \max_{i \in \{1, \ldots, c_t\}} \boldsymbol{h}'^\top \boldsymbol{C}_{\cdot, i}, \tag{7}$$

where $\boldsymbol{C}_{\cdot, i}$ denotes the $i$-th column of modulated prototypes.

**Adaptive Regularization**: Due to the inherent heterogeneity of different tasks, a fixed penalty coefficients $\lambda$ will cause suboptimal performance. For adaptive regularization, vanilla $\lambda$ selection via grid search and cross-validation incurs prohibitive computational costs of $\mathcal{O}(lm^3)$ for $l$ candidates where the expanded feature dimension $m$ is extremely large (McDonnell et al., 2023). To achieve our efficiency desideratum, we draw inspiration from an adaptive Generalized Cross-Validation (GCV) (Golub et al., 1979) framework that analytically approximates cross-validation error without explicit validation steps that require calculating large matrix inverses.

Given new task data $\boldsymbol{H}_t \in \mathbb{R}^{n_t \times m}$, we first obtain its singular value decomposition (SVD) as $\boldsymbol{H}_t = \boldsymbol{U}_t \boldsymbol{\Sigma}_t \boldsymbol{V}_t^\top$, where $\boldsymbol{U}_t \in \mathbb{R}^{n_t \times r}$ and $\boldsymbol{V}_t \in \mathbb{R}^{m \times r}$ are semi-orthogonal column matrices that satisfy $\boldsymbol{U}_t^\top \boldsymbol{U}_t = \boldsymbol{I}_r$, $\boldsymbol{V}_t^\top \boldsymbol{V}_t = \boldsymbol{I}_r$, and $\boldsymbol{\Sigma}_t = \text{diag}(s_1, \ldots, s_r) \in \mathbb{R}^{r \times r}$ contains non-zero singular values with $r = \text{rank}(\boldsymbol{H}_t) = \min(n_t, m)$ (typically full rank due to numerical precision). The SVD time complexity is $\mathcal{O}(n_t r m)$. For $l$ candidate regularization coefficients $\lambda \in \Lambda = \{\lambda_{min}, \ldots, \lambda_{max}\}$ on a log scale, we use the following steps to compute the GCV criterion for each one:

First, in $\mathcal{O}(lr)$ time, we get the shrinkage matrix and calculate the effective degrees-of-freedom by:

$$\boldsymbol{D}_t = \frac{\boldsymbol{\Sigma}_t^2}{\boldsymbol{\Sigma}_t^2 + \lambda \boldsymbol{I}_r}, \quad \text{df}(\lambda) = \text{tr}(\boldsymbol{D}_t) = \sum_{i=1}^{r} \frac{s_i^2}{s_i^2 + \lambda}. \tag{8}$$

Then, we reconstruct the prediction value of ridge regression in $\mathcal{O}(l n_t r c_t)$ time by

$$\hat{\boldsymbol{Y}}_t = \boldsymbol{U}_t(\text{vecdiag}(\boldsymbol{D}_t) \otimes \boldsymbol{1}_c^\top) \odot \boldsymbol{U}_t^\top \boldsymbol{Y}_t, \tag{9}$$

where $\otimes$ denotes the outer product, $\odot$ denotes the Hadamard product, and $\text{vecdiag}$ extracts the diagonal elements into a column vector. Finally, we obtain the GCV value by

$$\text{GCV}(\lambda) = \frac{\|\boldsymbol{Y}_t - \hat{\boldsymbol{Y}}_t(\lambda)\|_F^2}{n_t \left(1 - \frac{\text{df}(\lambda)}{n_t}\right)^2}, \tag{10}$$

with time complexity being $\mathcal{O}(l n_t c_t)$. The optimal regularization parameter is then selected by:

$$\lambda_t^* = \arg \min_{\lambda \in \Lambda} \text{GCV}(\lambda). \tag{11}$$

Considering the projected dimension $m$ is extremely large, we make the mild assumption that $m > n_t$, and $l c_t \ll m$, thus $r = \min(n_t, m) = n_t$. The original $l$ loop complexity is $\mathcal{O}(l n_t r c_t) = \mathcal{O}(l n_t^2 c_t) \ll \mathcal{O}(n_t^2 m) = \mathcal{O}(n_t r m)$. Hence, the time complexity is determined by SVD, at $\mathcal{O}(n_t^2 m)$. Compared to vanilla cross-validation taking $\mathcal{O}(lm^3)$, the time consumption is greatly reduced.

**Accelerated Prototype Calculation**: Upon determining the optimal regularization parameter $\lambda$ through GCV, we solve Eq. 6 to obtain class prototypes $\boldsymbol{C}_t$. While vanilla matrix inversion via LU decomposition provides a baseline implementation, for the sake of computational efficiency, we exploit the inherent positive-definiteness of $\boldsymbol{G}_t + \lambda_t \boldsymbol{I}_t$ to achieve computational acceleration through Cholesky factorization by:

$$\boldsymbol{L}_t \boldsymbol{L}_t^\top = \boldsymbol{G}_t + \lambda_t^* \boldsymbol{I}_m, \qquad \boldsymbol{C}_t = \boldsymbol{L}_t^{-\top}(\boldsymbol{L}_t^{-1} \boldsymbol{S}_t), \tag{12}$$

where $\boldsymbol{L}_t$ denotes the lower-triangular Cholesky factor. This approach reduces theoretical complexity from $\mathcal{O}(\frac{2}{3} m^3)$ to $\mathcal{O}(\frac{1}{3} m^3)$ for factorization, with triangular solves requiring half the FLOPs of general linear system solutions. The numerical stability of this method is ensured by the condition number bound $\kappa(\boldsymbol{L}_t) \leq \kappa(\boldsymbol{G}_t + \lambda_t^* \boldsymbol{I}_m)$, making it particularly suitable for ill-conditioned streaming scenarios where $\boldsymbol{G}_t$ may accumulate numerical noise over tasks. The decorrelation effect of the streaming ridge classification is visualized via the transformation from Figure 3(b) to (c).

Table 1: **Performance Comparison on Pre-trained ViT-B/16 Models.** We report the average training time per task ($\tau_{\text{train}}$), average post-extraction training time ($\tau_{\text{post}}$), and overall accuracy ($\bar{A}$) across three benchmark datasets: CIFAR-100, CUB-200-2011, and VTAB.

| Method | CIFAR-100 | | | CUB-200-2011 | | | VTAB | | |
|---|---|---|---|---|---|---|---|---|---|
| | $\tau_{\text{train}}(\downarrow)$ | $\tau_{\text{post}}(\downarrow)$ | $\bar{A}(\uparrow)$ | $\tau_{\text{train}}(\downarrow)$ | $\tau_{\text{post}}(\downarrow)$ | $\bar{A}(\uparrow)$ | $\tau_{\text{train}}(\downarrow)$ | $\tau_{\text{post}}(\downarrow)$ | $\bar{A}(\uparrow)$ |
| L2P | $263.54_{\pm0.10}$ | $183.56_{\pm0.36}$ | $87.74_{\pm0.46}$ | $52.02_{\pm0.04}$ | $37.03_{\pm0.07}$ | $77.48_{\pm1.43}$ | $42.10_{\pm0.04}$ | $36.45_{\pm0.02}$ | $81.24_{\pm0.67}$ |
| DualPrompt | $231.89_{\pm0.63}$ | $153.66_{\pm0.47}$ | $87.47_{\pm0.58}$ | $46.28_{\pm0.05}$ | $31.64_{\pm0.06}$ | $79.89_{\pm1.44}$ | $38.14_{\pm0.16}$ | $31.64_{\pm0.06}$ | $80.85_{\pm1.34}$ |
| InfLoRA | $220.82_{\pm0.44}$ | $140.31_{\pm0.41}$ | $91.10_{\pm0.36}$ | $45.31_{\pm0.19}$ | $30.97_{\pm0.22}$ | $80.65_{\pm0.73}$ | $35.80_{\pm0.28}$ | $29.26_{\pm0.17}$ | $88.73_{\pm0.57}$ |
| SEMA | $241.60_{\pm0.82}$ | $160.87_{\pm0.69}$ | $92.04_{\pm0.25}$ | $48.21_{\pm0.23}$ | $32.96_{\pm0.20}$ | $84.31_{\pm0.37}$ | $45.50_{\pm0.13}$ | $39.82_{\pm0.26}$ | $91.18_{\pm0.46}$ |
| MoE-Adapter | $187.91_{\pm0.61}$ | $106.37_{\pm0.53}$ | $90.43_{\pm0.46}$ | $37.89_{\pm0.15}$ | $22.61_{\pm0.12}$ | $79.65_{\pm0.32}$ | $32.28_{\pm0.19}$ | $26.06_{\pm0.14}$ | $86.39_{\pm0.77}$ |
| EASE | $621.26_{\pm1.05}$ | $583.4_{\pm0.76}$ | $92.96_{\pm0.25}$ | $138.78_{\pm0.36}$ | $122.48_{\pm0.27}$ | $89.56_{\pm0.43}$ | $108.35_{\pm0.97}$ | $94.73_{\pm1.11}$ | $94.02_{\pm0.15}$ |
| RanPAC | $98.62_{\pm0.42}$ | $84.42_{\pm0.44}$ | $\mathbf{94.21}_{\pm0.11}$ | $37.95_{\pm0.15}$ | $33.83_{\pm0.13}$ | $92.67_{\pm0.20}$ | $63.68_{\pm0.18}$ | $61.44_{\pm0.17}$ | $94.16_{\pm0.32}$ |
| F-OAL | $71.05_{\pm0.23}$ | $57.13_{\pm0.19}$ | $91.96_{\pm0.29}$ | $6.24_{\pm0.09}$ | $2.04_{\pm0.05}$ | $91.13_{\pm0.15}$ | $3.19_{\pm0.05}$ | $1.03_{\pm0.02}$ | $94.68_{\pm0.32}$ |
| **Fly-CL** | $\mathbf{19.07}_{\pm0.07}$ | $\mathbf{5.38}_{\pm0.01}$ | $93.89_{\pm0.12}$ | $\mathbf{4.43}_{\pm0.11}$ | $\mathbf{0.35}_{\pm0.01}$ | $\mathbf{93.84}_{\pm0.18}$ | $\mathbf{2.48}_{\pm0.13}$ | $\mathbf{0.34}_{\pm0.03}$ | $\mathbf{96.54}_{\pm0.38}$ |

# 5 EXPERIMENTS

## 5.1 EXPERIMENTAL SETUP

**Datasets and Backbones:** We conduct experiments using various architectures, including transformer-based and CNN-based backbones. Specifically, we utilize the Vision Transformer (ViT-B/16) (Dosovitskiy et al., 2020) and ResNet-50 (He et al., 2016) as representative architectures. We test our method on five widely used datasets: CIFAR-100 (Krizhevsky et al., 2009), CUB-200-2011 (Wah et al., 2011), VTAB (Zhai et al., 2019), ImageNet-R (Hendrycks et al., 2021a), and ImageNet-A (Hendrycks et al., 2021b). Further details of the data setup are provided in Appendix E.

**Baselines:** We compare Fly-CL against eight baselines, including two prompt-based approaches: L2P (Wang et al., 2022b) and DualPrompt (Wang et al., 2022a), three lora/adapter-based approaches: InfLoRA (Liang & Li, 2024), SEMA (Wang et al., 2025a), and MoE-Adapter (Yu et al., 2024), as well as three representation-based methods: EASE (Zhou et al., 2024), RanPAC (McDonnell et al., 2023), and F-OAL (Zhuang et al., 2024). In Fly-CL, we find that applying data normalization according to the specific combination of backbone and dataset is beneficial; a detailed analysis can be found in Appendix C.3. For a fair comparison, all baselines use the same data normalization strategy as Fly-CL. Comparisons among the different baselines and implementation details are provided in Appendices D and E, respectively.

**Evaluation Metrics:** To assess CL performance, we employ four metrics: average accuracy ($A_t$), last stage accuracy ($A_T$), backward transfer ($BWT$), and overall accuracy ($\bar{A}$). The average accuracy at stage $t$ is defined as: $A_t = \frac{1}{t}\sum_{i=1}^{t} a_{t,i}$, where $a_{t,i}$ denotes the test accuracy on the $i$-th task after training on the $t$-th task. Specially, we refer to the average accuracy at the last stage $T$ as the last stage accuracy ($A_T$). The backward transfer is denoted as $BWT = \frac{1}{t-1}\sum_{i=1}^{T}(a_{t,i} - a_{i,i})$. The overall accuracy is computed as the mean of $A_t$ across all $T$ tasks: $\bar{A} = \frac{1}{T}\sum_{i=1}^{T} A_t$. To evaluate computational efficiency, we introduce two time-related metrics: average training time per task ($\tau_{\text{train}}$) and average post-extraction training time ($\tau_{\text{post}}$). Here, $\tau_{\text{train}}$ represents the total training time amortized across all tasks, while $\tau_{\text{post}}$ is derived by subtracting the average feature extraction time for each task (using the pre-trained model) from $\tau_{\text{train}}$. $\tau_{\text{post}}$ is a more precise metric for evaluating algorithm-specific time consumption, as it excludes the shared preprocessing overhead. [1]

## 5.2 LOW LATENCY AND HIGH ACCURACY

The main CL results across various datasets, architectures, and task settings are summarized in Tables 1, 2, and 5. Our framework's key strength is achieving CL accuracy comparable to or exceeding SOTA performance with significantly lower computational costs, as measured by both $\tau_{\text{train}}$ and $\tau_{\text{post}}$. In Table 1, using ViT-B/16, Fly-CL reduces $\tau_{\text{post}}$ by 91% on CIFAR-100 with only a marginal accuracy drop of 0.32% compared to SOTA methods. On CUB-200-2011 and VTAB, Fly-CL achieves 83% and 67% reductions in $\tau_{\text{post}}$ versus the most efficient baseline while improving overall accuracy by

---

[1] $\tau_{\text{train}}$ and $\tau_{\text{post}}$ are measured in seconds (wall clock time); $A_t$ and $\bar{A}$ are measured in %.

Table 2: **Performance Comparison on Pre-trained ResNet-50 Models.** We report the average training time per task ($\tau_{\text{train}}$), average post-extraction training time ($\tau_{\text{post}}$), and overall accuracy ($\bar{A}$) across three benchmark datasets: CIFAR-100, CUB-200-2011, and VTAB. The best results are highlighted in **bold**.

| Method | CIFAR-100 | | | CUB-200-2011 | | | VTAB | | |
|---|---|---|---|---|---|---|---|---|---|
| | $\tau_{\text{train}}(\downarrow)$ | $\tau_{\text{post}}(\downarrow)$ | $\bar{A}(\uparrow)$ | $\tau_{\text{train}}(\downarrow)$ | $\tau_{\text{post}}(\downarrow)$ | $\bar{A}(\uparrow)$ | $\tau_{\text{train}}(\downarrow)$ | $\tau_{\text{post}}(\downarrow)$ | $\bar{A}(\uparrow)$ |
| RanPAC | $55.68_{\pm0.97}$ | $46.65_{\pm0.88}$ | $82.72_{\pm0.22}$ | $58.74_{\pm0.84}$ | $54.35_{\pm0.99}$ | $78.72_{\pm0.40}$ | $50.15_{\pm0.36}$ | $47.94_{\pm0.38}$ | $92.80_{\pm0.40}$ |
| F-OAL | $80.74_{\pm0.35}$ | $71.78_{\pm0.35}$ | $66.63_{\pm0.71}$ | $5.19_{\pm0.09}$ | $1.69_{\pm0.01}$ | $60.84_{\pm1.67}$ | $2.76_{\pm0.03}$ | $0.55_{\pm0.01}$ | $26.15_{\pm2.50}$ |
| **Fly-CL** | $\mathbf{14.28_{\pm0.04}}$ | $\mathbf{5.25_{\pm0.01}}$ | $\mathbf{84.61_{\pm0.16}}$ | $\mathbf{3.90_{\pm0.31}}$ | $\mathbf{0.44_{\pm0.08}}$ | $\mathbf{80.25_{\pm0.10}}$ | $\mathbf{2.53_{\pm0.10}}$ | $\mathbf{0.34_{\pm0.02}}$ | $\mathbf{94.00_{\pm0.15}}$ |

Table 3: **Performance Comparison on Pre-trained ViT-B/16 Models using Online Learning Setting.** $^\circ$ denotes methods in online mode. We report the average training time per task ($\tau_{\text{train}}$), average post-extraction training time ($\tau_{\text{post}}$), and overall accuracy ($\bar{A}$) across three benchmark datasets: CIFAR-100, CUB-200-2011, and VTAB. The best results are highlighted in **bold**.

| Method | CIFAR-100 | | | CUB-200-2011 | | | VTAB | | |
|---|---|---|---|---|---|---|---|---|---|
| | $\tau_{\text{train}}(\downarrow)$ | $\tau_{\text{post}}(\downarrow)$ | $\bar{A}(\uparrow)$ | $\tau_{\text{train}}(\downarrow)$ | $\tau_{\text{post}}(\downarrow)$ | $\bar{A}(\uparrow)$ | $\tau_{\text{train}}(\downarrow)$ | $\tau_{\text{post}}(\downarrow)$ | $\bar{A}(\uparrow)$ |
| RanPAC$^\circ$ | $1236.74_{\pm1.36}$ | $1223.56_{\pm1.07}$ | $92.48_{\pm0.31}$ | $242.54_{\pm1.56}$ | $238.36_{\pm1.47}$ | $91.89_{\pm0.26}$ | $122.89_{\pm0.46}$ | $120.69_{\pm0.42}$ | $93.41_{\pm0.57}$ |
| F-OAL$^\circ$ | $164.58_{\pm0.71}$ | $151.27_{\pm0.64}$ | $91.48_{\pm0.42}$ | $31.34_{\pm0.32}$ | $27.20_{\pm0.28}$ | $91.60_{\pm0.22}$ | $11.49_{\pm0.14}$ | $9.47_{\pm0.16}$ | $95.28_{\pm0.21}$ |
| **Fly-CL$^\circ$** | $\mathbf{25.46_{\pm0.32}}$ | $\mathbf{12.57_{\pm0.26}}$ | $\mathbf{92.96_{\pm0.14}}$ | $\mathbf{6.44_{\pm0.08}}$ | $\mathbf{2.33_{\pm0.05}}$ | $\mathbf{92.59_{\pm0.13}}$ | $\mathbf{3.17_{\pm0.05}}$ | $\mathbf{1.09_{\pm0.04}}$ | $\mathbf{96.38_{\pm0.24}}$ |

1.17% and 2.38% over the best-performing methods, respectively. In Table 2, with ResNet-50, Fly-CL improves overall accuracy by 1.89%, 1.53%, and 1.20% on CIFAR-100, CUB-200-2011, and VTAB, respectively, while reducing $\tau_{\text{post}}$ by 93%, 74%, and 38% versus the most efficient baselines. These improvements align with transformer-based backbone trends. Notably, F-OAL exhibits significant performance degradation on CNN backbones, presumably due to error accumulation in its iterative update mechanism, but Fly-CL does not suffer from this issue. These results highlight Fly-CL's ability to balance computational efficiency and accuracy across diverse CL scenarios, demonstrating its robustness. Results on datasets with severe domain shifts are presented in Table 6.

Additionally, the time difference between $\tau_{\text{train}}$ and $\tau_{\text{post}}$ in Tables 1 and 2 indicates that feature extraction becomes the dominant time consumer in Fly-CL. For a fair comparison with the baselines, we do not apply additional acceleration techniques here. However, in practical applications, techniques like model quantization (e.g., INT8) can further reduce feature extraction time by around 4× without significant accuracy degradation, thereby enhancing the speedup ratio. For hardware-specific deployment, frameworks like TVM (Chen et al., 2018) can be utilized to maximize efficiency.

Furthermore, Fly-CL can be easily adapted to Online CL setups by updating the $G$ and $S$ matrices and solving Eq. 6 for each batch, without concatenating all batch embeddings within a task. The results in Table 3 indicate that batch-mode Fly-CL remains superior to other baselines in training time and is also competitive in accuracy.

### 5.3 FACTORS CONTRIBUTING TO COMPUTATIONAL SPEEDUP

Our analysis in Sections 4.1 and 4.2 demonstrates that the proposed framework achieves significant speedup over the vanilla implementation through component-level optimizations. To quantify these improvements precisely, we split the post-extraction training time into three key components (as illustrated in Figure 2) and evaluate Fly-CL against its vanilla implementation under the CUB-200-2011 setting in Table 4. The components include: (1) Random Projection: Acceleration via weight sparsity induced by sparse projection versus the dense version. (2) Ridge Selection: Time reduction achieved by GCV, which eliminates the need for explicit cross-validation. (3) Prototype Calculation: Optimization from LU decomposition to Cholesky factorization. Additionally, the inference stage also benefits from the activation sparsity induced by the top-$k$ operation in similarity comparisons.

### 5.4 ABLATION STUDY AND HYPERPARAMETER SENSITIVITY ANALYSIS

Using ViT-B/16 as the backbone, we conduct ablation studies by individually removing the projection layer (w/o proj), the streaming ridge classification (w/o ridge), and the data normalization components

Table 4: **Time Savings for Post-Extracting Components on CUB-200-2011.** We compare the theoretical time complexity per task ($T_{\text{theory}}$) and the actual runtime per task ($T_{\text{actual}}$) for each component on an NVIDIA GeForce RTX 3090 GPU. The optimized implementations demonstrate significant speedups across all components.

| Method | Random Projection | | Ridge Selection | | Prototype Calculation | | Similarity Comparison | |
|---|---|---|---|---|---|---|---|---|
| | $T_{\text{theory}}$ | $T_{\text{actual}}$ | $T_{\text{theory}}$ | $T_{\text{actual}}$ | $T_{\text{theory}}$ | $T_{\text{actual}}$ | $T_{\text{theory}}$ | $T_{\text{actual}}$ |
| vanilla | $\mathcal{O}(mn_t d)$ | $0.22\pm0.03$ | $\mathcal{O}(lm^3)$ | $7.34\pm0.12$ | $\mathcal{O}(\frac{2}{3}m^3)$ | $0.20\pm0.01$ | $\mathcal{O}(mn_t c_t)$ | $0.21\pm0.01$ |
| optimized | $\mathcal{O}(mn_t p)$ | $\mathbf{0.08\pm0.02}$ | $\mathcal{O}(mn_t^2)$ | $\mathbf{0.14\pm0.01}$ | $\mathcal{O}(\frac{1}{3}m^3)$ | $\mathbf{0.10\pm0.01}$ | $\mathcal{O}(kn_t c_t)$ | $\mathbf{0.08\pm0.01}$ |

(w/o norm). The results in Figure 4 demonstrate that each component contributes significantly to overall performance. Removing any of these components, or all of them (w/o all), leads to noticeable performance degradation.

We also analyze the sensitivity of the key hyperparameters in Fly-CL: $m$ (projection dimension), $p$ (weight sparsity), and $k$ (activation sparsity) in Figure 5. Increasing $m$ improves accuracy, with performance saturating beyond $m = 10,000$. Notably, Fly-CL does not suffer from the curse of dimensionality, which can be attributed to the fact that random projection into a higher-dimensional space preserves pairwise distances between data points, as guaranteed by the Johnson-Lindenstrauss Lemma (Johnson et al., 1984). CL performance increases monotonically with $p$, and no significant performance drop occurs as long as $p$ does not take an excessively small value. A sufficiently large $k$ value avoids information loss, while a smaller value suppresses noisy dimensions. Thus, finding an appropriate trade-off can lead to optimal accuracy. Encouragingly, Figure 5(c) shows a broad plateau for optimal $k$ selection. Based on our empirical results, we set $m = 10,000$, $p = 300$, and $k = 3,000$ as default values.

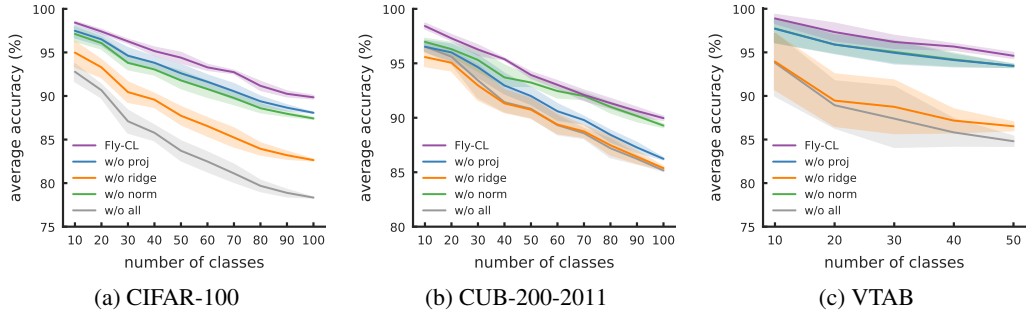

(a) CIFAR-100       (b) CUB-200-2011       (c) VTAB

Figure 4: **Accuracy Curves from Ablation Studies on Three Datasets.** We report average accuracy ($A_t$) for each stage. w/o refers to the removal of the specific component.

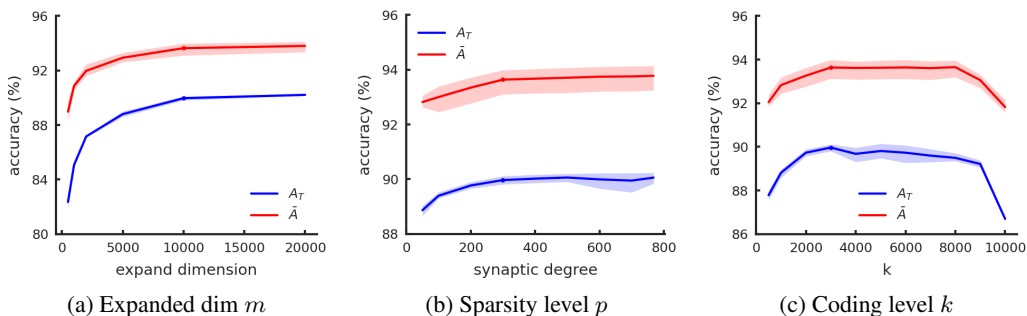

(a) Expanded dim $m$     (b) Sparsity level $p$     (c) Coding level $k$

Figure 5: **Sensitivity Analysis for Expanded dim $m$, Weight Sparsity $p$, and Activation Sparsity $k$ on CUB-200-2011.** We report average accuracy in last task ($A_T$) and overall accuracy ($\bar{A}$). The dots denote the default values we use across experiments.

# 6 DISCUSSION: CONSISTENCY BETWEEN RIDGE CLASSIFICATION AND HEBBIAN LEARNING RULES UNDER CONTINUAL LEARNING

In this section, we demonstrate that a biologically plausible local learning rule, constructed from Hebbian and anti-Hebbian plasticity (Hebb, 2005), converges to the same stationary point as ridge classification. This equivalence provides a theoretical bridge between the synaptic dynamics in the KC→MBON pathway and the streaming ridge classification algorithm employed in Fly-CL.

We begin by defining a synaptic update rule that depends only on variables locally available at each synapse. Let $h' \in \mathbb{R}^m$ denote the pre-synaptic activity vector and $\hat{y} = C_t^\top h'$ the post-synaptic response predicted by the current weights $C_t \in \mathbb{R}^{m \times c_t}$. The synaptic update is defined as:

$$\Delta C = C_{t+1} - C_t = \eta\big(h'y^\top - h'\hat{y}^\top - \lambda C_t\big), \tag{13}$$

where $\eta > 0$ is a small learning rate, $y \in \mathbb{R}^{c_t}$ is the one-hot ground-truth label. This rule consists of three biologically meaningful components:

1. **Hebbian term $h'y^\top$**, which strengthens synapses when both pre-synaptic activity and the target post-synaptic signal co-activate.
2. **Anti-Hebbian term $-h'\hat{y}^\top$**, which suppresses correlations between the input and the model's own prediction, implementing an error-correcting mechanism.
3. **Weight decay $-\lambda C_t$**, corresponding to metabolic cost or homeostatic constraints.

This formulation is biologically plausible: it requires only pre-synaptic activity, post-synaptic activity, and a modulatory teaching signal, constituting a standard three-factor learning rule commonly observed in neuromodulated plasticity.

Taking the expectation over data samples $(h', y)$, we obtain:

$$\mathbb{E}[\Delta C] = \eta\big(\mathbb{E}[h'y^\top] - \mathbb{E}[h'\hat{y}^\top] - \lambda C_t\big) = \eta\big(S_t - G_t C_t - \lambda C_t\big), \tag{14}$$

where we define

$$G_t = \mathbb{E}[h'h'^\top] \in \mathbb{R}^{m \times m}, \qquad S_t = \mathbb{E}[h'y^\top] \in \mathbb{R}^{m \times c_t}.$$

Thus, the expected synaptic dynamics follow the linear recursion:

$$C_{t+1} = C_t + \eta\big(S_t - (G_t + \lambda I_m)C_t\big). \tag{15}$$

Taking the continuous-time limit $\eta \to 0$ yields the ordinary differential equation:

$$\frac{dC_t}{dt} = \kappa\big(S_t - (G_t + \lambda I_m)C_t\big), \tag{16}$$

where $\kappa > 0$ rescales time. The stationary solution satisfies:

$$(G_t + \lambda I_m)C_t^\star = S_t,$$

and therefore:

$$C_t^\star = (G_t + \lambda I_m)^{-1}S_t, \tag{17}$$

which is exactly the closed-form solution of ridge regression.

In our representation-based CIL setting, since all samples in the same task are available simultaneously, we do not need to integrate the continuous-time dynamics in Eq. 16. Instead, we can directly compute the ridge solution to obtain the optimal stationary classifier.

# 7 CONCLUSION

In this work, inspired by the decorrelation mechanism in the fly olfactory circuit, we propose an efficient CL framework, Fly-CL. Fly-CL significantly reduces computational overhead during training while achieving competitive performance compared to SOTA methods. This framework integrates several key components: data normalization, feature extraction, sparse random projection with top-$k$ operation, and streaming ridge classification, each contributing to the overall efficiency and effectiveness of the system. Ultimately, this work demonstrates that neurobiological principles, particularly sparse coding and progressive decorrelation, can effectively address the efficiency-accuracy trade-off in artificial continual learning systems.

ACKNOWLEDGMENTS

This work was supported by the Fundamental and Interdisciplinary Disciplines Breakthrough Plan of the Ministry of Education of China (JYB2025XDXM503).

ETHICS STATEMENT

This research strictly complies with the ICLR Code of Ethics. Our empirical evaluations are conducted exclusively on standard, publicly accessible benchmarks, entirely free of personal, sensitive, or human-derived information. All data usage respects the original licenses, and we foresee no negative societal impacts regarding privacy, security, or algorithmic bias. Furthermore, by advancing efficient continual learning, Fly-CL contributes to sustainable and green AI development by significantly reducing the computational footprint and energy consumption typically required for retraining models from scratch.

REPRODUCIBILITY STATEMENT

The theoretical proofs for our theorems are provided in Appendix B. Full details regarding the experimental setup, covering datasets, benchmarks, model configurations, evaluation metrics, and hyperparameters, are outlined in Section 5 and Appendix E. To ensure reproducibility, we rely exclusively on publicly accessible datasets and have made the complete source code available.

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

# A   ALGORITHM PSEUDOCODE

---

**Algorithm 1** Fly-CL Training Pipeline

---

**Input:** Sequentially arriving data $\mathcal{D}_t = \{(\boldsymbol{x}_i^t, y_i^t)\}_{i=1}^{n_t}$ where $t = 1, \ldots, T$. Pre-trained encoder $f_\theta$. Projection operator $Z(\cdot) : \mathbb{R}^d \to \mathbb{R}^m$. Penalty coefficient candidates $\Lambda = \{\lambda_1, \ldots, \lambda_l\}$. Zero-initialized matrices $\boldsymbol{G}_0 \in \mathbb{R}^{m \times m}$ and $\boldsymbol{S}_0 \in \mathbb{R}^{m \times c_t}$.

**Output:** Modulated Prototypes $\boldsymbol{C} \in \mathbb{R}^{m \times c_t}$.

1: **for** $t = 1, \ldots, T$ **do**
2:      $r = \min(n_t, m)$
3:      Get compressed embedding for each datum $\boldsymbol{v}_i^t = f_\theta(\boldsymbol{x}_i^t) \in \mathbb{R}^d$
4:      Transform to high-dim sparse embedding $Z(\boldsymbol{v}_i^t) = \text{top-}k\,(\boldsymbol{W}\boldsymbol{v}_i^t)$         $\triangleright \mathcal{O}(mn_t p)$
5:      Concatenate $Z(\boldsymbol{v}_i^t)$ to get $\boldsymbol{H}_t \in \mathbb{R}^{n_t \times m}$
6:      $\boldsymbol{G}_t \leftarrow \boldsymbol{G}_{t-1} + \boldsymbol{H}_t^\top \boldsymbol{H}_t$         $\triangleright \mathcal{O}(n_t m^2)$
7:      $\boldsymbol{S}_t \leftarrow \boldsymbol{S}_{t-1} + \boldsymbol{H}_t^\top \boldsymbol{Y}_t$         $\triangleright \mathcal{O}(n_t c_t m)$
8:      $\boldsymbol{U}_t, \boldsymbol{\Sigma}_t, \boldsymbol{V}_t = \text{svd}(\boldsymbol{H}_t)$         $\triangleright \mathcal{O}(n_t r m)$
9:      **for** $\lambda \in \Lambda$ **do**
10:          $\boldsymbol{D}_t = \frac{\boldsymbol{\Sigma}_t^2}{\boldsymbol{\Sigma}_t^2 + \lambda \boldsymbol{I}_r}$         $\triangleright \mathcal{O}(lr)$
11:          $\text{df}(\lambda) = \text{tr}(\boldsymbol{D}_t) = \sum_{i=1}^r \frac{s_i^2}{s_i^2 + \lambda}$         $\triangleright \mathcal{O}(lr)$
12:          $\hat{\boldsymbol{Y}}_t = \boldsymbol{U}_t(\text{vecdiag}(\boldsymbol{D}_t) \otimes \mathbf{1}_c^\top) \odot \boldsymbol{U}_t^\top \boldsymbol{Y}_t$         $\triangleright \mathcal{O}(ln_t r c_t)$
13:          $\text{GCV}(\lambda) = \frac{\|\boldsymbol{Y}_t - \hat{\boldsymbol{Y}}_t(\lambda)\|_F^2}{n_t\left(1 - \frac{\text{df}(\lambda)}{n_t}\right)^2}$         $\triangleright \mathcal{O}(ln_t c_t)$
14:      **end for**
15:      Select $\lambda_t^* = \arg\min_{\lambda \in \Lambda} \text{GCV}(\lambda)$
16:      $\boldsymbol{L}_t \boldsymbol{L}_t^\top = \boldsymbol{G}_t + \lambda_t^* \boldsymbol{I}_m$         $\triangleright \mathcal{O}(\frac{1}{3}m^3)$
17:      $\boldsymbol{C}_t = \boldsymbol{L}_t^{-\top}(\boldsymbol{L}_t^{-1} \boldsymbol{S}_t)$         $\triangleright \mathcal{O}(m^2 c_t)$
18: **end for**

---

**Algorithm 2** Fly-CL Inference Pipeline

---

**Input:** Sequentially arriving data $\mathcal{D}_t = \{(\boldsymbol{x}_i^t, y_i^t)\}_{i=1}^{n_t}$ where $t = 1, \ldots, T$. Pre-trained encoder $f_\theta$. Projection operator $Z$. Modulated class prototypes $\boldsymbol{C}_t \in \mathbb{R}^{m \times c_t}$.

**Output:** Predicted labels $\hat{y}$.

1: Get compressed embedding for each datum $\boldsymbol{v} = f_\theta(\boldsymbol{x}) \in \mathbb{R}^d$
2: Transform to high-dim sparse embedding $Z(\boldsymbol{v}) = \text{top-}k\,(\boldsymbol{W}\boldsymbol{v})$         $\triangleright \mathcal{O}(mp)$
3: Compute prediction $\hat{y} = Z(\boldsymbol{v})^\top \boldsymbol{C}_t$         $\triangleright \mathcal{O}(kc_t)$

---

# B   COMPLETE THEORETICAL ANALYSIS

In this section, we present a comprehensive theoretical analysis of the sparsification effects on both the weights and activations in the random projection operation.

## B.1   INFORMATION PRESERVING FOR SPARSE CONNECTIONS IN RANDOM PROJECTION MATRIX

A common approach to demonstrate that sparse random matrix multiplication preserves information equivalently to its dense counterpart lies in proving the matrix's near-preservation of full column rank. For our sparse random matrix $\boldsymbol{W} \in \mathbb{R}^{m \times d}$ where $m > d$, we prove that $\boldsymbol{W}$ almost surely maintains rank $d$.

**Theorem B.1.** *Given the matrix $\boldsymbol{W} \in \mathbb{R}^{m \times d}$, where $m > d$, with each row having exactly $p$ non-zero entries, which are randomly sampled from $\mathcal{N}(0, 1)$. Let $\mathcal{W} \in \mathbb{R}^{d \times d}$ be any square submatrix of $\boldsymbol{W}$. Then, for any $\epsilon > 0$, it holds that*

$$\mathbb{P}\left(|\det(\mathcal{W})| \geq \left(\frac{p}{d}\right)^{d/2} \sqrt{d!}\exp(-d^{1/2+\epsilon})\right) = 1 - o(1).$$

*Thus, for sufficiently large $p$ and $d$, any submatrix $\mathcal{W}$ is invertible with probability at least $1 - o(1)$.*

*Proof.* According to aforementioned definition, we have $\mathbb{E}(\boldsymbol{W}_{ij}) = 0$ and $\text{Var}(\boldsymbol{W}_{ij}) = \frac{p}{d}$. Considering $\boldsymbol{R} = \frac{1}{\sigma}\mathcal{W}$, which satisfies $\mathbb{E}(\boldsymbol{R}_{ij}) = 0$, $\text{Var}(\boldsymbol{R}_{ij}) = 1$, we can conclude, based on (Tao & Vu, 2006, Theorem 8.9), that

$$\mathbb{P}\left(|\det(\boldsymbol{R})| \geq \sqrt{d!}\exp(-d^{1/2+\epsilon})\right) = 1 - o(1). \tag{18}$$

By using $\sigma = \sqrt{\frac{p}{d}}$ and $\det(\boldsymbol{R}) = \sigma^{-d}\det(\mathcal{W})$ for substitution, we complete the proof. $\square$

To better demonstrate the information preserving property of our construction, another line of validation is to utilize the random matrix's distance preservation property, which is theoretically guaranteed by the Johnson-Lindenstrauss (JL) lemma (Johnson et al., 1984). Specifically, we conduct an empirical simulation to verify that the normalization of our sparse projection matrix $\boldsymbol{\Phi} = \sqrt{\frac{d}{mp}}\boldsymbol{W}$ preserves pairwise Euclidean distances between feature vectors with high probability. For every pair of vectors $(\boldsymbol{x}_1, \boldsymbol{x}_2)$, we calculate the Distortion Ratio defined as $\text{Ratio} = \frac{\|\boldsymbol{\Phi}(\boldsymbol{x}_1-\boldsymbol{x}_2)\|_2^2}{\|\boldsymbol{x}_1-\boldsymbol{x}_2\|_2^2}$. We randomly select 500 data points from the extracted features of CIFAR-100 dataset and compute the pairwise distances between all point pairs, with the results illustrated in Figure 6. By analyzing the distribution of these ratios across all vector pairs, the resulting histogram shows a strong

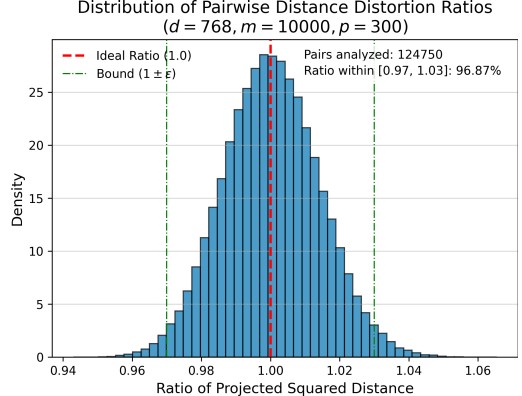

Figure 6: Empirical Verification of the Johnson-Lindenstrauss (JL) Property using Sparse Random Projection.

concentration around the ideal value of 1. If we set $\epsilon$ to 0.03, then the vast majority of the ratios are empirically confined within the bounds of $[1 - \epsilon, 1 + \epsilon]$. This concentration strongly validates that our sparse projection structure-comprising only $p$ non-zero $\mathcal{N}(0, 1)$ entries per row-effectively maintains the geometric structure of the high-dimensional data.

## B.2 ROBUSTNESS OF TOP-$k$ SPARSIFICATION ON HIGH-DIMENSIONAL EMBEDDINGS

Let the high-dimensional embedding vector be $\boldsymbol{h} \in \mathbb{R}^m$. After applying the top-$k$ operation, we obtain a sparsified vector $\boldsymbol{h}' \in \mathbb{R}^m$, where only the $k$ largest absolute values in $\boldsymbol{h}$ are retained, and the remaining elements are set to zero. We aim to prove that when $k = \Omega(m^{\alpha})$ (with $0 < \alpha < 1$, i.e., not overly sparse), the performance degradation is negligible.

According to statistic learning theory (Bartlett & Mendelson, 2002) and extreme value theory (Coles et al., 2001; Fisher & Tippett, 1928), we start with the following two widely-accepted assumption:

**Assumption B.2.** *Assume that the "energy" (squared $\ell_2$-norm) of the embedding vector $x$ is concentrated in a few dimensions, i.e., there exists a constant $C > 0$ such that:*

$$\mathbb{E}\left[\frac{\sum_{i=1}^{k}\boldsymbol{h}_i^2}{\|\boldsymbol{h}\|_2^2}\right] \geq 1 - \frac{C}{k},$$

*where where $\boldsymbol{h}_i$ denotes the $i$-th largest value in $\boldsymbol{h}$.*

**Assumption B.3.** *Assume that the performance loss function $L(\boldsymbol{h}, y)$ of the downstream task (e.g., classifier) is Lipschitz continuous with respect to input perturbations, i.e., there exists a constant $M > 0$ such that:*

$$|L(\boldsymbol{h}, y) - L(\boldsymbol{h}', y)| \leq M \cdot \|\boldsymbol{h} - \boldsymbol{h}'\|.$$

Regarding the approximation error of top-$k$ operation, we show it is bounded.

**Theorem B.4.** *Under the Assumption B.2, the sparsification error satisfies:*

$$E\left[\|\boldsymbol{h} - \boldsymbol{h}'\|_2^2\right] \leq \frac{C}{k} \cdot \mathbb{E}\left[\|\boldsymbol{h}\|_2^2\right].$$

*Proof.* By Assumption B.2:

$$\mathbb{E}\left[\sum_{i=k+1}^{d} \boldsymbol{h}_i^2\right] \leq \frac{C}{k} \cdot \mathbb{E}\left[\|\boldsymbol{h}\|_2^2\right].$$

Thus,

$$\mathbb{E}\left[\|\boldsymbol{h} - \boldsymbol{h}'\|_2^2\right] = \mathbb{E}\left[\sum_{i=k+1}^{m} \boldsymbol{h}_i^2\right] \leq \frac{C}{k} \cdot \mathbb{E}\left[\|\boldsymbol{h}\|_2^2\right].$$

$\square$

Then, we can quantify the upper bound of possible performance degradation.

**Theorem B.5.** *Under Assumption B.3, the performance degradation due to sparsification satisfies:*

$$\mathbb{E}\left[|L(\boldsymbol{h}, y) - L(\boldsymbol{h}', y)|\right] \leq M \cdot \sqrt{\frac{C}{k} \cdot \mathbb{E}[\|\boldsymbol{h}\|_2^2]}$$

*Proof.* By the Cauchy-Schwarz inequality and Theorem B.4, we can derive that

$$\mathbb{E}\left[|L(\boldsymbol{h}, y) - L(\boldsymbol{h}', y)|\right] \leq M \cdot \mathbb{E}\left[\|\boldsymbol{h} - \boldsymbol{h}'\|_2\right]$$
$$\leq M \cdot \sqrt{\mathbb{E}\left[\|\boldsymbol{h} - \boldsymbol{h}'\|_2^2\right]}$$
$$\leq M \cdot \sqrt{\frac{C}{k} \cdot \mathbb{E}\left[\|\boldsymbol{h}\|_2^2\right]}.$$

$\square$

From Theorem B.5, we establish in Theorem B.6 that moderate sparsity does not result in significant performance degradation, as the error decreases exponentially with the increasing expanded dimension $m$.

**Theorem B.6.** *For top-$k$ sparsification in the expanded dimension $m$, the performance degradation is bounded by:*

$$\mathbb{E}\left[|L(\boldsymbol{h}, y) - L(\boldsymbol{h}', y)|\right] \leq M \cdot \sqrt{\frac{C}{k} \cdot \mathbb{E}[\|\boldsymbol{h}\|_2^2]},$$

*where $L(\cdot)$ is a performance loss function for downstream tasks and $C$, $M$ are constants. To ensure negligible performance degradation, we require:*

$$\sqrt{\frac{C}{k} \cdot \mathbb{E}[\|\boldsymbol{h}\|_2^2]} \leq \mathcal{O}\left(\frac{1}{\sqrt{m^\alpha}}\right),$$

*i.e., when $k = \Omega(m^\alpha)$ ($0 < \alpha < 1$), the error bound decays exponentially with increasing dimension.*

For example:

- If $k = \Omega(d^{0.5})$, the performance degradation is $\mathcal{O}(d^{-0.25})$.
- If $k = \Omega(d^{0.8})$, the performance degradation is $\mathcal{O}(d^{-0.4})$.

## C    ADDITIONAL RESULTS

### C.1    EXPERIMENTS IN LONGER TASK SEQUENCES

To evaluate the long-term stability of Fly-CL, we conduct experiments with task sequences twice as long as those in Table 1. Results are presented in Table 5 and Figure 7. Overall, both $\tau_{\text{train}}$ and $\tau_{\text{post}}$ are shorter than those in Table 1 due to fewer samples per task. Fly-CL improves overall accuracy by $0.54\%$, $1.21\%$, and $1.58\%$ compared to SOTA methods, while significantly reducing average post-extraction training time by $89\%$, $74\%$, and $59\%$ compared to the most efficient baselines. These results are consistent with the trends observed in Table 1 and Figure 8, demonstrating the robustness of Fly-CL across different task lengths.

Table 5: **Performance Comparison on Pre-trained ViT-B/16 Models with Longer Task Sequence.** We report the average training time per task ($\tau_{\text{train}}$), average post-extraction training time ($\tau_{\text{post}}$), and overall accuracy ($\bar{A}$) across three benchmark datasets: CIFAR-100, CUB-200-2011, and VTAB. The best results are highlighted in **bold**.

| Method | CIFAR-100 | | | CUB-200-2011 | | | VTAB | | |
|---|---|---|---|---|---|---|---|---|---|
| | $\tau_{\text{train}}(\downarrow)$ | $\tau_{\text{post}}(\downarrow)$ | $\bar{A}(\uparrow)$ | $\tau_{\text{train}}(\downarrow)$ | $\tau_{\text{post}}(\downarrow)$ | $\bar{A}(\uparrow)$ | $\tau_{\text{train}}(\downarrow)$ | $\tau_{\text{post}}(\downarrow)$ | $\bar{A}(\uparrow)$ |
| L2P | $147.47_{\pm0.36}$ | $107.59_{\pm0.26}$ | $82.69_{\pm0.91}$ | $30.93_{\pm0.25}$ | $23.39_{\pm0.26}$ | $72.83_{\pm1.45}$ | $32.17_{\pm0.39}$ | $29.31_{\pm0.38}$ | $71.84_{\pm1.42}$ |
| Dualprompt | $130.12_{\pm0.09}$ | $91.05_{\pm0.07}$ | $83.42_{\pm0.86}$ | $27.66_{\pm0.18}$ | $20.28_{\pm0.18}$ | $77.93_{\pm0.91}$ | $29.44_{\pm0.24}$ | $26.62_{\pm0.23}$ | $78.46_{\pm1.14}$ |
| InfLoRA | $124.80_{\pm0.39}$ | $84.36_{\pm0.12}$ | $88.18_{\pm0.34}$ | $27.12_{\pm0.14}$ | $19.85_{\pm0.09}$ | $75.67_{\pm0.16}$ | $29.71_{\pm0.21}$ | $26.92_{\pm0.17}$ | $81.09_{\pm0.73}$ |
| SEMA | $128.95_{\pm0.46}$ | $88.57_{\pm0.19}$ | $88.93_{\pm0.52}$ | $27.98_{\pm0.25}$ | $20.52_{\pm0.18}$ | $80.82_{\pm0.11}$ | $30.36_{\pm0.27}$ | $27.55_{\pm0.20}$ | $84.56_{\pm0.48}$ |
| MoE-Adapter | $132.67_{\pm0.62}$ | $92.31_{\pm0.45}$ | $88.42_{\pm0.28}$ | $30.25_{\pm0.19}$ | $22.82_{\pm0.15}$ | $78.62_{\pm0.35}$ | $29.25_{\pm0.18}$ | $26.81_{\pm0.13}$ | $83.24_{\pm0.57}$ |
| EASE | $350.92_{\pm0.39}$ | $331.76_{\pm0.29}$ | $90.14_{\pm0.51}$ | $83.38_{\pm0.22}$ | $75.05_{\pm0.16}$ | $91.47_{\pm0.65}$ | $64.68_{\pm0.73}$ | $57.24_{\pm0.62}$ | $90.26_{\pm0.41}$ |
| RanPAC | $44.53_{\pm0.58}$ | $37.03_{\pm0.59}$ | $93.68_{\pm0.24}$ | $20.68_{\pm0.32}$ | $18.18_{\pm0.31}$ | $92.65_{\pm0.26}$ | $41.49_{\pm0.74}$ | $40.02_{\pm0.59}$ | $93.62_{\pm0.32}$ |
| F-OAL | $16.32_{\pm0.02}$ | $8.91_{\pm0.02}$ | $92.63_{\pm0.46}$ | $3.65_{\pm0.02}$ | $1.10_{\pm0.01}$ | $91.48_{\pm0.23}$ | $2.17_{\pm0.16}$ | $0.71_{\pm0.03}$ | $94.96_{\pm0.27}$ |
| **Fly-CL** | $\mathbf{8.31_{\pm0.04}}$ | $\mathbf{1.00_{\pm0.01}}$ | $\mathbf{94.22_{\pm0.09}}$ | $\mathbf{2.76_{\pm0.06}}$ | $\mathbf{0.29_{\pm0.01}}$ | $\mathbf{93.86_{\pm0.27}}$ | $\mathbf{1.70_{\pm0.10}}$ | $\mathbf{0.29_{\pm0.02}}$ | $\mathbf{96.54_{\pm0.38}}$ |

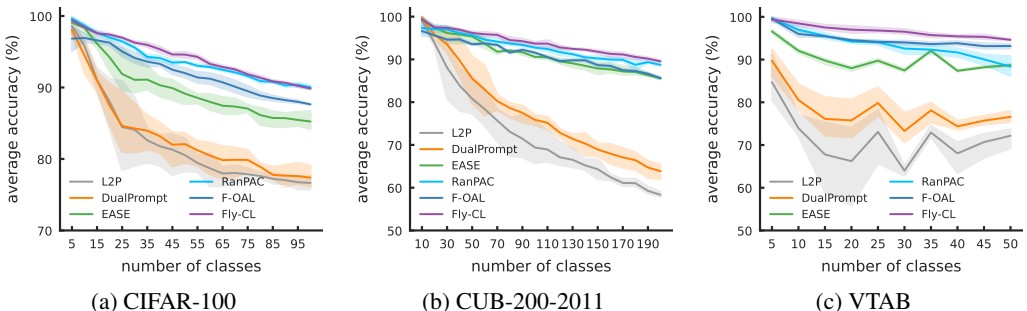

(a) CIFAR-100          (b) CUB-200-2011          (c) VTAB

Figure 7: **Accuracy Curves of Different Methods on Pre-trained ViT-B/16 with Longer Task Sequence.** The average accuracy ($A_t$) is reported for each dataset. These results align with and extend the quantitative analysis presented in Table 5.

### C.2    EXPERIMENTS ON DATASETS WITH SEVERE DOMAIN SHIFT

Table 6 summarizes the results on ImageNet-R and ImageNet-A under severe domain shift. Compared with existing continual learning baselines, Fly-CL achieves the best overall accuracy on both datasets ($83.19\%$ on ImageNet-R and $67.98\%$ on ImageNet-A), comparable with the previous SOTA RanPAC. More importantly, Fly-CL attains these improvements with substantially lower computation cost. Its average training time per task is reduced by an order of magnitude compared to prompt-based methods (e.g., L2P, DualPrompt) and much faster than EASE, while its post-extraction training time is almost negligible ($0.21$s vs. $67.71$s for RanPAC on ImageNet-R). These results demonstrate that Fly-CL is not only robust to severe distribution shifts but also highly efficient, making it especially suitable for practical continual learning scenarios where both accuracy and efficiency are critical.

Table 6: **Performance Comparison on Pre-trained ViT-B/16 Models with Severe Domain Shift.** We report the average training time per task ($\tau_{\text{train}}$), average post-extraction training time ($\tau_{\text{post}}$), and overall accuracy ($\bar{A}$) across two benchmark datasets: ImageNet-R and ImageNet-A. The best results are highlighted in **bold**.

| Method | ImageNet-R | | | ImageNet-A | | |
|---|---|---|---|---|---|---|
| | $\tau_{\text{train}}(\downarrow)$ | $\tau_{\text{post}}(\downarrow)$ | $\bar{A}(\uparrow)$ | $\tau_{\text{train}}(\downarrow)$ | $\tau_{\text{post}}(\downarrow)$ | $\bar{A}(\uparrow)$ |
| L2P | $131.97_{\pm0.46}$ | $110.56_{\pm0.42}$ | $76.13_{\pm0.21}$ | $56.28_{\pm0.32}$ | $48.92_{\pm0.27}$ | $48.86_{\pm0.08}$ |
| Dualprompt | $117.80_{\pm0.34}$ | $96.58_{\pm0.30}$ | $73.92_{\pm0.46}$ | $49.60_{\pm0.28}$ | $42.31_{\pm0.26}$ | $57.05_{\pm0.13}$ |
| InfLoRA | $62.32_{\pm0.33}$ | $41.05_{\pm0.23}$ | $82.15_{\pm0.41}$ | $25.71_{\pm0.30}$ | $18.50_{\pm0.24}$ | $62.32_{\pm0.28}$ |
| SEMA | $68.85_{\pm0.21}$ | $47.34_{\pm0.19}$ | $81.89_{\pm0.17}$ | $28.65_{\pm0.26}$ | $21.23_{\pm0.22}$ | $61.79_{\pm0.32}$ |
| MoE-Adapter | $66.37_{\pm0.29}$ | $45.02_{\pm0.24}$ | $81.76_{\pm0.33}$ | $26.27_{\pm0.41}$ | $18.89_{\pm0.37}$ | $61.72_{\pm0.21}$ |
| EASE | $311.00_{\pm0.29}$ | $274.36_{\pm0.25}$ | $81.69_{\pm0.24}$ | $80.80_{\pm0.19}$ | $73.47_{\pm0.22}$ | $65.03_{\pm0.28}$ |
| RanPAC | $76.25_{\pm0.35}$ | $67.71_{\pm0.28}$ | $83.02_{\pm0.12}$ | $32.43_{\pm0.13}$ | $28.86_{\pm0.11}$ | $67.28_{\pm0.09}$ |
| F-OAL | $16.51_{\pm0.11}$ | $8.80_{\pm0.04}$ | $80.62_{\pm0.25}$ | $3.99_{\pm0.07}$ | $1.05_{\pm0.02}$ | $63.99_{\pm0.30}$ |
| **Fly-CL** | $\mathbf{7.55_{\pm0.04}}$ | $\mathbf{0.21_{\pm0.02}}$ | $\mathbf{83.19_{\pm0.14}}$ | $\mathbf{3.10_{\pm0.03}}$ | $\mathbf{0.15_{\pm0.01}}$ | $\mathbf{67.98_{\pm0.17}}$ |

### C.3 DATA NORMALIZATION STRATEGY

While data normalization is a well-established technique for improving classification performance in i.i.d. scenarios, its effectiveness in facilitating CL with frozen pre-trained encoders remains unclear. Our results indicate that applying proper architecture-specific normalization to input images significantly improves the learning performance compared to baseline CL methods (Table 7). The optimal normalization strategies for the included backbones differ. Across all tested datasets, ViT-B/16 (Dosovitskiy et al., 2020) benefits more from standard normalization that projects inputs into the $[-1, 1]$ range, while ResNet-50 (He et al., 2016) achieves optimal performance when normalized using ImageNet statistics.

We hypothesize that the imporved performance arises from a reduced feature distribution shift across tasks. Proper normalization preserves the geometry of the pre-trained feature manifold, which is crucial for prototype-based classification, where cosine similarity measures depend on the angular relationships between features. Our empirical results suggest that input normalization may serve as a fundamental defense against forgetting by anchoring the feature space topology to the original pre-training distribution.

Table 7: **Comparison of CL Performance across Pre-trained Models and Normalization Strategies.** We report overall accuracy ($\bar{A}$). Normalization methods includes: "None" (no data normalization), "ImageNet" (ImageNet statistics), and "Standard"(scaled to the $[-1, 1]$).

| Backbone | CIFAR | | | CUB | | | VTAB | | |
|---|---|---|---|---|---|---|---|---|---|
| | None | ImageNet | Standard | None | ImageNet | Standard | None | ImageNet | Standard |
| ViT-B/16 | $91.64_{\pm0.62}$ | $87.87_{\pm0.62}$ | $\mathbf{93.89_{\pm0.12}}$ | $93.04_{\pm0.37}$ | $90.68_{\pm0.42}$ | $\mathbf{93.84_{\pm0.18}}$ | $95.26_{\pm0.68}$ | $95.47_{\pm0.52}$ | $\mathbf{96.54_{\pm0.38}}$ |
| ResNet-50 | $80.66_{\pm0.48}$ | $\mathbf{84.61_{\pm0.16}}$ | $83.09_{\pm0.48}$ | $75.08_{\pm1.23}$ | $\mathbf{80.25_{\pm0.10}}$ | $76.78_{\pm1.08}$ | $92.45_{\pm0.71}$ | $\mathbf{94.00_{\pm0.15}}$ | $92.76_{\pm0.54}$ |

### C.4 MEMORY CONSUMPTION

We also compare the memory consumption of Fly-CL against other methods in Table 8 using ViT-B/16 with the same task sequence as in Table 1. For fairness, we use a batch size of 128 across all methods and datasets. The results show Fly-CL also has the minimal memory cost, strengthening the efficiency of our method.

### C.5 MEMORY-TIME TRADEOFF IN HIGH-DIM PROJECTIONS

We present the trade-off between memory, training time, and overall accuracy in Table 9. The overall accuracy gradually saturates as the dimension increases, while memory and training time grow quadratically. Therefore, we chose 10,000 as the dimension in our simulations. As long as the

Table 8: **Memory Usage (GB) of Different Methods on Pre-trained ViT-B/16.** We report highest peak memory usage of each methods. The best results are highlighted in **bold**.

| Method | CIFAR-100 | CUB-200-2011 | VTAB |
|---|---|---|---|
| L2P | 16.4GB | 16.4GB | 16.4GB |
| DualPrompt | 13.6GB | 13.6GB | 13.6GB |
| InfLoRA | 14.1GB | 14.1GB | 14.1GB |
| SEMA | 12.9GB | 12.9GB | 12.9GB |
| MoE-Adapter | 20.2GB | 20.2GB | 20.2GB |
| EASE | 12.2GB | 12.2GB | 12.2GB |
| RanPAC | 12.2GB | 12.2GB | 22.8GB |
| F-OAL | 12.2GB | 4.9GB | 4.5GB |
| **Fly-CL** | **6.7GB** | **4.6GB** | **4.3GB** |

dimension does not exceed 10,000, both memory and training time consumption remain lower than those of previous methods, as summarized in Table 1.

Table 9: **Memory-Time-Accuracy comparison with increasing projection dimension.** It's conducted on the CUB dataset using ViT B/16. We report highest peak memory usage of each methods. The best results are highlighted in **bold**.

| Dimension | 1000 | 2000 | 5000 | 10000 | 20000 |
|---|---|---|---|---|---|
| Memory | 2.8G | 2.8G | 3.0G | 4.6G | 9.3G |
| $\tau_{train}$ | $4.19_{\pm 0.02}$ | $4.20_{\pm 0.05}$ | $4.25_{\pm 0.07}$ | $4.43_{\pm 0.11}$ | $5.13_{\pm 0.08}$ |
| $\bar{A}$ | $90.87_{\pm 0.49}$ | $91.97_{\pm 0.52}$ | $92.93_{\pm 0.41}$ | $93.84_{\pm 0.18}$ | $93.90_{\pm 0.52}$ |

## C.6 ADDITIONAL VISUALIZED FIGURES AND EVALUATION METRIC DURING THE TRAINING PROCESS

Here, we present a more detailed breakdown of the training processes for ViT-B/16 and ResNet-50. Results from Tables 1 and 2 are visualized in Figures 8 and 9. We list the average accuracy of different methods at different stages across three datasets. We additionally report the last-stage accuracy ($A_T$)

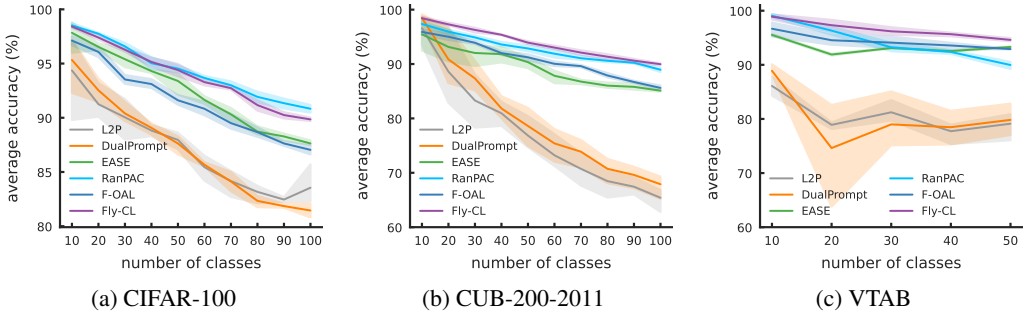

(a) CIFAR-100        (b) CUB-200-2011        (c) VTAB

Figure 8: **Accuracy Curves of Different Methods on Pre-trained ViT-B/16.** The average accuracy ($A_t$) is reported for each dataset. These results align with and extend the quantitative analysis presented in Table 1.

and the backward transfer score (a representative forgetting metric) of Table 1 in Table 10 to provide a more comprehensive evaluation.

## C.7 ADDITIONAL SENSITIVITY ANALYSIS ACROSS TASK COMPLEXITY, NUMBER OF TASKS/CLASSES, AND DIFFERENT PRETRAINED BACKBONES

We further conduct additional sensitivity analyses under three complementary settings: (i) ImageNet-A with 10 tasks and 20 classes per task using ViT-B/16 to examine the effect of task complexity

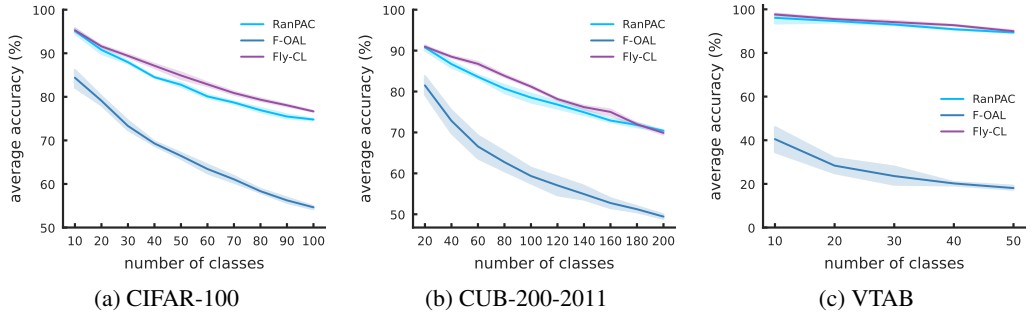

(a) CIFAR-100    (b) CUB-200-2011    (c) VTAB

Figure 9: **Accuracy Curves of Different Methods on Pre-trained ResNet-50.** The average accuracy ($A_t$) is reported for each dataset. These results align with and extend the quantitative analysis presented in Table 2.

Table 10: **Performance Comparison on Pre-trained ViT-B/16 Models.** We report the last stage accuracy ($A_T$), and backward transfer ($BWT$) across three benchmark datasets: CIFAR-100, CUB-200-2011, and VTAB.

| Method | CIFAR-100 | | CUB-200-2011 | | VTAB | |
|---|---|---|---|---|---|---|
| | $A_T(\uparrow)$ | $BWT(\uparrow)$ | $A_T(\uparrow)$ | $BWT(\uparrow)$ | $A_T(\uparrow)$ | $BWT(\uparrow)$ |
| L2P | $83.55_{\pm1.53}$ | $-6.23_{\pm0.41}$ | $65.41_{\pm1.84}$ | $-13.14_{\pm0.57}$ | $79.12_{\pm2.18}$ | $-8.04_{\pm0.78}$ |
| DualPrompt | $81.45_{\pm0.52}$ | $-7.06_{\pm0.20}$ | $67.90_{\pm1.89}$ | $-12.12_{\pm0.53}$ | $79.83_{\pm2.37}$ | $-7.63_{\pm0.75}$ |
| InfLoRA | $86.56_{\pm0.46}$ | $-5.07_{\pm0.16}$ | $69.45_{\pm0.56}$ | $-11.53_{\pm0.25}$ | $87.88_{\pm0.73}$ | $-4.59_{\pm0.24}$ |
| SEMA | $87.47_{\pm0.43}$ | $-4.70_{\pm0.13}$ | $73.66_{\pm0.36}$ | $-9.94_{\pm0.19}$ | $89.28_{\pm0.60}$ | $-4.06_{\pm0.22}$ |
| MoE-Adapter | $86.88_{\pm0.32}$ | $-4.97_{\pm0.15}$ | $68.11_{\pm0.41}$ | $-12.11_{\pm0.23}$ | $88.06_{\pm0.48}$ | $-4.51_{\pm0.19}$ |
| EASE | $87.63_{\pm0.20}$ | $-4.66_{\pm0.06}$ | $85.10_{\pm0.19}$ | $-5.68_{\pm0.08}$ | $93.30_{\pm0.07}$ | $-2.48_{\pm0.04}$ |
| RanPAC | $\mathbf{90.83_{\pm0.41}}$ | $\mathbf{-3.47_{\pm0.11}}$ | $88.95_{\pm0.48}$ | $-4.16_{\pm0.22}$ | $89.97_{\pm0.71}$ | $-3.79_{\pm0.26}$ |
| F-OAL | $87.04_{\pm0.50}$ | $-4.90_{\pm0.14}$ | $85.61_{\pm0.50}$ | $-5.43_{\pm0.24}$ | $92.91_{\pm0.07}$ | $-2.66_{\pm0.04}$ |
| **Fly-CL** | $89.85_{\pm0.17}$ | $-3.82_{\pm0.09}$ | $\mathbf{89.97_{\pm0.18}}$ | $\mathbf{-3.80_{\pm0.07}}$ | $\mathbf{94.61_{\pm0.35}}$ | $\mathbf{-2.01_{\pm0.15}}$ |

(Figure 10); (ii) CUB-200-2011 with 20 tasks and 10 classes per task using ViT-B/16 to evaluate the impact of a larger number of tasks and classes (Figure 11); and (iii) CUB-200-2011 with 10 tasks and 20 classes per task using ResNet-50 to assess the influence of different backbone architectures (Figure 12). Across all settings, the observed trends remain consistent with those reported in Figure 5.

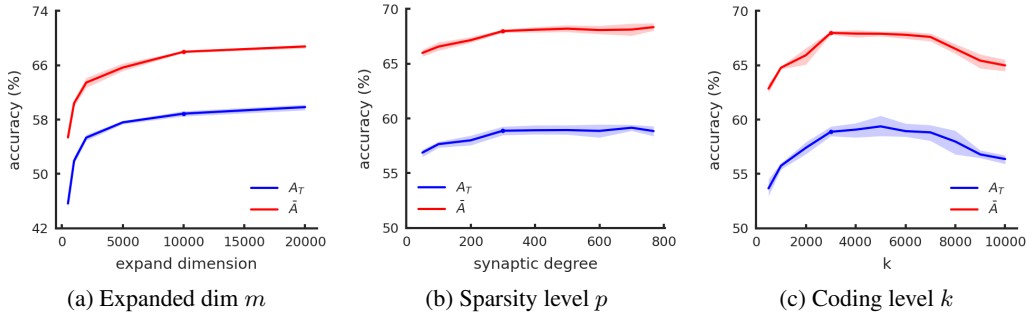

(a) Expanded dim $m$    (b) Sparsity level $p$    (c) Coding level $k$

Figure 10: **Sensitivity Analysis for Expanded dim $m$, Weight Sparsity $p$, and Activation Sparsity $k$ on ImageNet-A.** We report average accuracy in last task ($A_T$) and overall accuracy ($\bar{A}$). The dots denote the default values we use across experiments.

## C.8 ADDITIONAL EXPERIMENTS ON LARGER PRE-TRAINED MODELS

In our previous experiments, we primarily adopted ResNet-50 and ViT-B/16 as backbones for feature extraction. To further evaluate the scalability of Fly-CL on modern foundation models, we employ the vision encoder of Qwen2.5-VL-7B (Bai et al., 2025) to extract visual features. As shown in Table 11, increasing the scale of the pre-trained backbone consistently leads to improved performance.

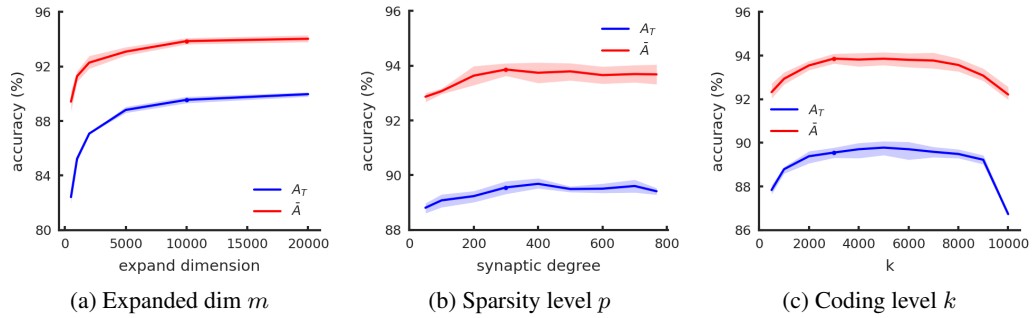

(a) Expanded dim $m$     (b) Sparsity level $p$     (c) Coding level $k$

Figure 11: **Sensitivity Analysis for Expanded dim $m$, Weight Sparsity $p$, and Activation Sparsity $k$ on CUB-200 with Longer Task Sequence.** We report average accuracy in last task ($A_T$) and overall accuracy ($\bar{A}$). The dots denote the default values we use across experiments.

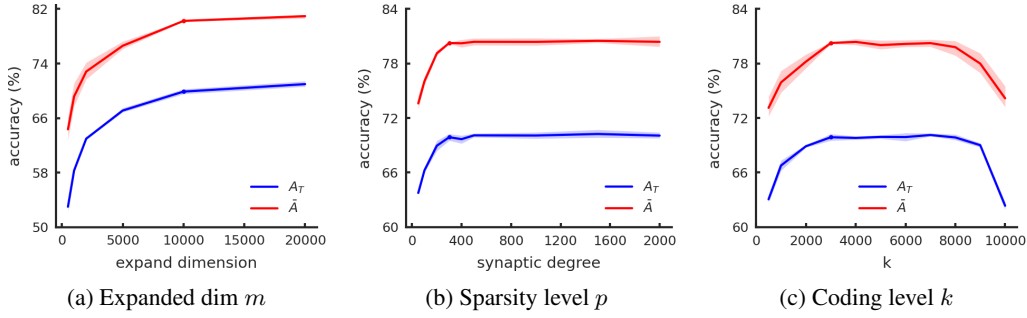

(a) Expanded dim $m$     (b) Sparsity level $p$     (c) Coding level $k$

Figure 12: **Sensitivity Analysis for Expanded dim $m$, Weight Sparsity $p$, and Activation Sparsity $k$ on CUB-200 with ResNet-50 as Backbone.** We report average accuracy in last task ($A_T$) and overall accuracy ($\bar{A}$). The dots denote the default values we use across experiments.

Furthermore, in Figure 13, we analyze the effect of expanding the projection dimension when using Qwen2.5-VL. The trend remains consistent with our earlier findings: performance improves steadily as the expanded dimension grows, but begins to saturate around $10,000$.

Table 11: **Performance Comparison on Different Scale Pretrained Models.** We report overall accuracy ($\bar{A}$) across three benchmark datasets: CIFAR-100, CUB-200-2011, and VTAB.

| Model | CIFAR-100 | CUB-200-2011 | VTAB |
|---|---|---|---|
| ResNet-50 | $84.61_{\pm0.16}$ | $80.25_{\pm0.10}$ | $94.00_{\pm0.15}$ |
| ViT-B/16 | $93.89_{\pm0.12}$ | $93.84_{\pm0.18}$ | $96.54_{\pm0.38}$ |
| Qwen2.5-VL-7B | $95.06_{\pm0.23}$ | $94.68_{\pm0.21}$ | $97.45_{\pm0.24}$ |

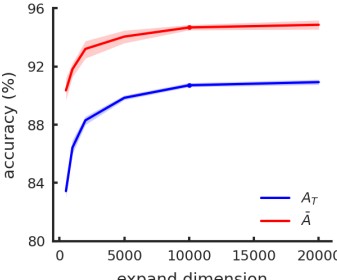

Figure 13: **Sensitivity Analysis for Expanded dim $m$ on CUB-200 with Qwen2.5-VL-7B as Backbone.**

# D   DETAILED DISCUSSION OF RELATED WORK

## D.1   COMPARISON WITH SEVERAL REPRESENTATION-BASED METHODS

We highlight the main advantages of our proposed Fly-CL over several related representation-based methods, including RanPAC (McDonnell et al., 2023), F-OAL (Zhuang et al., 2024), and RanDumb (Prabhu et al., 2024).

**Comparison with RanPAC.** RanPAC employs several Parameter-Efficient Transfer Learning (PETL) approach (Chen et al., 2022; Jia et al., 2022; Lian et al., 2022) to adapt the pre-trained model to the downstream domain in the first task, alongside a ridge classification with explicit cross-validation for all ridge candidates. Although effective, these two components make the entire pipeline computationally expensive (see Table 1, 2, and 5). In contrast, Fly-CL eliminates the need for PETL and significantly optimizes the ridge classification process. Additionally, we introduce a sparse projection layer with a top-$k$ operation, replacing the dense projection with ReLU, and analyze the impact of data normalization techniques. The speedup for each components can be refered to Table 4.

**Comparison with F-OAL.** F-OAL is originally designed for online CL and shares similarities with Fly-CL in feature extraction, random projection, and decorrelation. Although it can also be adapted to the CIL setting with batched data, it has several flaws under this circumstance. For instance, F-OAL lacks the top-$k$ operation to filter noisy components after random projection, and its iterative analytic classifier may accumulate errors, leading to significant performance degradation on ResNet-50 (see Table 2). Moreover, while F-OAL is efficient on CUB-200-2011 and VTAB, its computational cost scales more rapidly with sample size compared to Fly-CL, making it less efficient on CIFAR-100 (see Table 1, 2, and 5).

**Comparison with RanDumb.** RanDumb shares a similar pipeline with F-OAL and is also designed for online CL. Like F-OAL, it does not utilize a top-$k$-like operation, and its fixed penalty coefficient $\lambda$ may result in suboptimal performance. Crucially, RanDumb relies on StreamingLDA, which processes samples sequentially and cannot be parallelized for batch processing. This makes RanDumb significantly slower than all baselines evaluated in Table 1, 2, and 5.

## D.2   SUMMARY OF OTHER COMPARED BASELINES

**L2P** (Wang et al., 2022b) utilizes a prompt pool $\mathbf{P} = \{P_1, P_2, \cdots, P_M\}$ where $M$ is the size of the pool, to store task-specific knowledge. Each prompt $P_i$ is associated with a learnable key $K_i$ for key-value selection. By optimizing the cosine distance $\gamma(p(x), k_i)$, where $p(x)$ is the feature selected by the query function during the training process, L2P can select the most appropriate prompt to provide information that is specific to the task.

**DualPrompt** (Wang et al., 2022a) extends the key-value selection and optimization methods of L2P by further encoding different types of information into a task-invariant prompt $g$ and a task-specific prompt $e$. This is shown to be more effective in encoding the learned knowledge. It also decouples the higher-level prompt space by attaching prompts to different layers, which is crucial for the model to reduce forgetting and achieve effective knowledge sharing.

**EASE** (Zhou et al., 2024) first initializes and trains an adapter for each incoming task to encode task-specific information. It then extracts features of the current task and synthesizes prototypes of former classes to mitigate the subspace gaps between adapters. Finally, EASE constructs the full classifier and reweights the logits for prediction.

**InfLoRA** (Liang & Li, 2024) is a Interference-Free Low-Rank Adaptation method for continual learning. It injects a small set of parameters to constrain weight updates to a specific subspace. Critically, this subspace is designed to be orthogonal to the gradients of all past tasks while containing the gradient subspace of the new task, thereby eliminating interference and achieving an effective balance between model stability and plasticity.

**SEMA** (Wang et al., 2025a) introduces a self-expansion mechanism by dynamically adding modular adapters only when significant distribution shifts are detected. Each adapter consists of a functional module and a representation descriptor, which acts as a novelty detector to determine whether existing adapters can handle the new task. SEMA further maintains an expandable mixture router to compose adapters through weighted combination, enabling flexible reuse of old modules while expanding only on demand. This design achieves a sub-linear parameter growth rate while improving the stability–plasticity balance across tasks.

**MoE-Adapter** (Yu et al., 2024) enhances continual learning by injecting a mixture-of-experts adapter structure into transformer blocks. Each MoE-Adapter contains multiple expert adapters, and a learned routing network selects or mixes experts conditioned on the input. This architecture allows the model to capture diverse task-specific patterns while mitigating forgetting through expert specialization. By

leveraging the MoE structure, the method improves representational flexibility and achieves stronger adaptation capacity compared to using a single shared adapter.

### D.3 RELATIONSHIP WITH KANERVA'S SPARSE DISTRIBUTED MEMORY

Kanerva's Sparse Distributed Memory (SDM) (Kanerva, 1988) is a classical high-dimensional computing framework in which memory addresses are distributed in a large binary space, and read/write operations are performed by activating only those locations within a neighborhood defined by Hamming similarity. To highlight its conceptual connection to Fly-CL, we rewrite the SDM forward computation using the same notation as Fly-CL:

$$\hat{y} = \boldsymbol{C}^\top \big( f(\boldsymbol{W}\boldsymbol{v}) \big), \tag{19}$$

where $\boldsymbol{W}$ is an address transformation matrix, $\boldsymbol{C}$ is a content transformation matrix, and $f(\cdot)$ denotes a binary activation function.

Under this formulation, SDM and Fly-CL share two high-level principles: (i) expansion into a high-dimensional representational space, which promotes separability and reduces interference; and (ii) sparse activation, which suppresses noises and enhances discrimination through selective addressing.

Despite these conceptual parallels, Fly-CL differs from SDM in several important ways. First, $\boldsymbol{W}$ in SDM is updated through iterative read/write operations, whereas Fly-CL employs a fixed, randomly initialized projection matrix motivated by the fly olfactory circuit. Second, SDM does not incorporate any decorrelation mechanism analogous to the Fly-CL ridge-based KC-to-MBON transformation. Third, SDM operates in a binary space: the activation function $f(\cdot)$ produces binary addresses and similarity is evaluated via Hamming distance, while Fly-CL performs real-valued projections and uses cosine similarity for downstream matching and classification.

These distinctions illustrate that although Fly-CL and SDM share the broad philosophy of high-dimensional sparse representations, their architectural assumptions, objectives, and operating regimes differ substantially.

## E TRAINING DETAILS

### E.1 PRE-TRAINED MODELS

We use pre-trained ViT-B/16 and ResNet-50 models in our experiments. The ViT-B/16 checkpoint we used was first pretrained on ImageNet-21K and then fine-tuned on the ImageNet-1K dataset. All of which are loaded using the timm library. We list the dimensions of the extracted features and the download links for the checkpoints of each model in Table 12.

Table 12: **Information Related to the Pre-trained Models We Used in This Work.** We list the dimensions of the extracted features and provide corresponding download links for these pre-trained models.

| Model | feature dimension | Link |
|---|---|---|
| ViT-B/16 | 768 | Link |
| ResNet-50 | 2048 | Link |

### E.2 DATASETS

We evaluate our method on three benchmark datasets for CL tasks. Detailed information about these datasets, including download links, is provided in Table 13. For the experiments summarized in Tables 1, 2, and 3, we configure the number of training tasks as $T = 10$ for CIFAR-100 and CUB-200-2011, with 10 and 20 classes per task, respectively. For VTAB, we set $T = 5$ with 10 classes per task. In the longer task sequence experiments (Table 5), we double the task sequence length: for CIFAR-100 and CUB-200-2011, we set $T = 20$ with 5 and 10 classes per task, respectively, while for VTAB, we set $T = 10$ with 5 classes per task. For experiments in Table 6, we set $T = 10$ with 20 classes per task.

Table 13: **Details of CIFAR-100, CUB-200-2011, VTAB Datasets.** We list the number of training, validation samples and classes for the following datasets, along with the download links.

| Dataset | Training Samples | Validation Samples | Classes | Download Link |
|---|---|---|---|---|
| CIFAR-100 (Krizhevsky et al., 2009) | 50000 | 10000 | 100 | Link |
| CUB-200-2011 (Wah et al., 2011) | 9430 | 2358 | 200 | Link |
| VTAB (Zhai et al., 2019) | 1796 | 8619 | 50 | Link |
| Imagenet-R Hendrycks et al. (2021a) | 24000 | 6000 | 200 | Link |
| Imagenet-A Hendrycks et al. (2021b) | 5981 | 1519 | 200 | Link |

### E.3 EXPERIMENT SETUP

We reproduce the baseline results for L2P, DualPrompt, EASE, and RanPAC using the code provided by PILOT (Sun et al., 2023), ensuring that the learning parameters for each baseline align with the description in their original papers. For F-OAL, we adopt their official implementation for reproduction.

In our proposed Fly-CL, we set the expanded dimension $m$ to $10,000$, $p$ to $300$, and $k$ to $3,000$ across all experiments. For ViT-B/16, we apply standard data normalization, scaling each pixel value to the range $[-1, 1]$. For ResNet-50, we normalize the input images using ImageNet statistics. Given the prior knowledge of high multicollinearity in this task, we explore the penalty coefficient range starting from larger values, specifically from $10^6$ to $10^9$ on a log scale for ViT-B/16 and $10^4$ to $10^9$ for ResNet-50. Since prompt-based methods and PETL techniques are limited to transformer-based architectures, we compare Fly-CL only with RanPAC and F-OAL in the ResNet-50 setting. For RanPAC, we remove PETL and incorporate data normalization following their original implementation (McDonnell et al., 2023) for ResNet-50. All experiments are conducted using five different random seeds, and we report the mean $\pm$ standard deviation.

### E.4 ENVIRONMENTS

All experiments were conducted on a Linux server running Ubuntu 20.04.4 LTS, equipped with an Intel(R) Xeon(R) Platinum 8358P CPU at 2.60GHz and 8 NVIDIA GeForce RTX 3090 GPUs, using CUDA version 11.7. For model loading, we employed the timm library (version 0.9.16).

## F LIMITATIONS AND FUTURE WORK

Our proposed Fly-CL is theoretically applicable to various scenarios requiring feature separation. Its lightweight design further suggests potential utility in a wide range of Continual Learning and Metric Learning tasks. Looking forward, the framework can be broadened in several promising directions. First, incorporating large language model (LLM) priors (Qu et al., 2025a; Wang et al., 2024a; Qu et al., 2024) could enrich the initial state representations, providing more structured semantic knowledge to guide the decorrelation process. Second, our streaming feature separation mechanism naturally extends to deep reinforcement learning (RL) and RL post-training. By stabilizing representations, it can help mitigate out-of-distribution (OOD) shifts and improve generalization in offline RL (Mao et al., 2024a;b; 2023a;b; 2025; Shao et al., 2023; Qu et al., 2023), as well as accelerate the online RL fine-tuning of reasoning models (Qu et al., 2025c; Mao et al., 2026). Finally, combining Fly-CL with intelligent data sampling strategies (Wang et al., 2025b; Qu et al., 2025b; Zou et al., 2025a) could synergistically enhance training efficiency and adaptation robustness in complex, non-stationary environments.

Recent neuroscience research (Dasgupta et al., 2017) indicates that the random projection layer in the fly olfactory circuit may not be entirely random. Biological experiments also suggest the presence of certain constraints within this projection layer. Inspired by these findings, a promising direction for future research is to explore structuring the projection layer as an entity with learnable parameters, potentially enhancing its adaptability and performance.

## G   BROADER IMPACT

Our work provides a new perspective for enhancing the efficiency of CL using pre-trained models, which is crucial for real-world deployment, especially with increasingly large modern models. Fly-CL can help AI researchers and developers create more efficient CL algorithms.

On the other hand, the efficiency improvements in CL could potentially accelerate the development of AI systems that rapidly adapt to new domains without proper safeguards. This might lead to: (1) amplified propagation of biases present in sequential datasets, (2) reduced transparency as models continuously evolve beyond their initial training, and (3) potential misuse for generating tailored content at scale. We recommend implementing rigorous monitoring frameworks to track model behavior across learning phases.

## H   LLM USAGE DECLARATION

During the preparation of this manuscript, a large language model was employed exclusively for language refinement. Its role was limited to rephrasing certain passages and enhancing the overall clarity and readability of the text. All conceptual contributions, theoretical derivations, experimental design, and analysis were independently developed and verified by the authors. The LLM was not involved in generating research ideas, shaping methodologies, or producing novel scientific content. The authors bear full responsibility for the entirety of the paper.

