# OpenReview forum: "Fly-CL: A Fly-Inspired Framework for Enhancing Efficient Decorrelation and Reduced Training Time in Pre-trained Model-based Continual Representation Learning"
_ICLR.cc/2026/Conference — ICLR 2026 Poster_

### Official Review · Reviewer_awjr · 2025-10-30

**Soundness:** 3
**Presentation:** 3
**Contribution:** 3
**Rating:** 6
**Confidence:** 4

**Summary:**

The paper proposes FlyCL, which addresses multicollinearity in continual learning. They use a mixture of sparse random projection, top-k sparsity and streaming ridge classification. They demonstrate impressive computational speedups (90%), while maintaining comparable performance.

These are real and practical speedups, and this is definitely good engineering. The bio-inspiration however does not add much substance beyond motivation.

**Strengths:**

1. Very strong practical results, the paper shows very significant speedups with barely any loss in accuracy.
2. The method is clearly general enough to adapt to different architectures and datasets, in a plug-and-play manner.
3. The work has solid experimental evidence, good ablations, and statistical reporting.

**Weaknesses:**

1. The fly brain parallel is mostly surface-level (it looks like), and only inspires the sparse projection component.
2. Some hyperparameters seem to require architecture-specific tuning, although that does not necessarily negate the proposed generality.
3. Would be useful to examine if this scales to tasks at the scale of modern foundational models.

**Questions:**

1. How sensitive is the performance to m, p and k? The defaults chosen seem somewhat arbitrary.
2. When 10k dimensions are not sufficient, how does this scale to larger models?
3. Is there a benefit to keeping the projection matrix random as opposed to learning it?

---

> ### Author Response · Authors · 2025-11-27
> **Official Response to Reviewer awjr from Author (Part I)**
>
> We appreciate the reviewer's insightful comments. In response to the valuable feedback, we have conducted extensive new simulations to thoroughly address the concerns.
> > The fly brain parallel is mostly surface-level (it looks like), and only inspires the sparse projection component.
>
> We agree that these parallels could be articulated more clearly, and have revised the manuscript accordingly in the final version. Below, we provide a more detailed elaboration of these correspondences:
> 1. **Sparse Projection**: The sparse projection component in our model directly parallels the sparse connectivity between PNs and glomeruli in the fly olfactory circuit.
> 2. **Top-k Operation and APL Neuron Function**: The top-k operation in our model mirrors the winner-take-all mechanism implemented by APL neurons in the mushroom body. Specifically, APL neurons selectively retain the strongest KC activations, enforcing sparse coding—a feature our model replicates computationally.
> 3. **Layer Size Ratio and Biological Expansion**: The ratio of hidden-layer neurons to first-layer neurons in our model is based on the biological expansion observed between KCs and PN types in the fly olfactory circuit.
> 4. **Ridge Classification and Hebbian-like Learning**: For the final classification stage, we demonstrate that the parameter update rule from KCs to MBONs can be modeled as a biologically plausible Hebbian-like learning rule. Crucially, we prove that the ridge regression solution corresponds to the stationary point of this rule, enabling us to compute the solution directly rather than through iterative updates. The full derivation is provided in **Appendix B.3** for reference.
>
> These biological inspirations strengthen the interpretability and plausibility of our model. We thank the reviewer for prompting this clarification and have revised our manuscript accordingly.
>
> > Some hyperparameters seem to require architecture-specific tuning, although that does not necessarily negate the proposed generality.
>
> > How sensitive is the performance to m, p and k? The defaults chosen seem somewhat arbitrary.
>
> We sincerely appreciate the reviewer’s suggestion.
>
> In our initial submission, we included a sensitivity analysis on **CUB-200-2011** using **ViT-B/16**, evaluating performance across **10 tasks with 20 classes per task (Figure 5)**. To address the reviewer’s concerns more comprehensively, we have conducted additional hyperparameter sensitivity studies, detailed in **Appendix C.7 (Figures 10, 11, and 12)**. These new experiments explore:
>
> - Variations in **task complexity**,
> - Different **numbers of tasks/classes**, and
> - Performance across **multiple pretrained backbones**.
>
> Our results demonstrate **consistent robustness** across all experimental settings:
>
> - **Expanded dimension (m)** saturates at approximately **10,000**, beyond which further increases yield diminishing returns.
> - **Synaptic sparsity (p)** exhibits no performance degradation once it reaches **300**, indicating a reliable lower bound for model stability.
> - **Coding level (k)** shows a clear trend: accuracy initially improves with increasing k, peaks, and then gradually declines. The optimal range begins at **k ≈ 3000**, representing the lower bound for maximum accuracy.
>
> These findings reinforce the robustness of our approach under diverse conditions. We are happy to further discuss or expand on any of these points if needed.
>
> > Would be useful to examine if this scales to tasks at the scale of modern foundational models.
>
> > When 10k dimensions are not sufficient, how does this scale to larger models?
>
> We fully agree with the reviewer that employing modern foundation models as pretrained backbones would further strengthen our conclusions about **Fly-CL's scalability**.
>
> To address this point, we conducted an additional experiment using the **vision encoder of Qwen2.5-VL-7B** as the pretrained feature extractor (see **Appendix C.8, Table 11, and Figure 13**). Our results demonstrate that:
>
> - A 10,000-dimensional expansion is sufficient to achieve effective decorrelation with this larger model.
> - Further increasing the dimensionality (m) **provided no noticeable performance gains**, suggesting diminishing returns beyond this point.
>
> These findings reinforce that the choice of m is **inherently robust—there is no need for excessively large expansions, which would only introduce unnecessary memory overhead without tangible benefits**. We appreciate the opportunity to clarify this aspect and are happy to provide further details if needed.

---

> ### Author Response · Authors · 2025-11-27
> **Official Response to Reviewer awjr from Author (Part II)**
>
> > Is there a benefit to keeping the projection matrix random as opposed to learning it?
>
> In our framework, the projection matrix is randomly initialized and kept fixed during learning. Freezing this layer plays a crucial role in mitigating catastrophic forgetting, whereas making it learnable leads to significant interference in the projection parameters.
>
> To rigorously validate the importance of a fixed projection matrix, we introduced two additional baseline model for comparison. These baseline incorporates:
>
> 1. A large expansion layer (PNs → KCs),
> 2. Top-k sparse activation, and
> 3. Simple learnable linear layer (KCs → MBONs) instead of ridge regression to make PNs → KCs capable of learning.
>
> The only difference between the two variants of these new baseline is whether the expansion layer (PNs → KCs) is learnable or random.
>
> Our results below show that **using a fixed projection matrix** yields **significantly higher performance and drastically reduces forgetting** compared to the learnable version. All learnable parameters in both baselines were optimized via SGD with a learning rate of 0.01, following the same experimental settings as in Table 1. We report the overall accuracy $\bar{A}$.
>
> |  Method   | CIFAR-100  | CUB-200-2011 |    VTAB    |
> | :-------: | :--------: | :----------: | :--------: |
> |  random   | 61.81±1.33 |  73.09±1.79  | 67.79±2.76 |
> | learnable | 46.28±0.97 |  53.42±1.85  | 61.68±2.29 |

---

### Official Review · Reviewer_uEtz · 2025-10-31

**Soundness:** 3
**Presentation:** 2
**Contribution:** 3
**Rating:** 4
**Confidence:** 5

**Summary:**

This paper presents Fly-CL, an efficient framework for continual learning with frozen pre-trained encoders. It introduces two key components: (1) a sparse random projection with top-k activation sparsification to decorrelate features and improve prototype separability, and (2) a streaming ridge regression classifier with adaptive regularization via generalized cross-validation for stability and low computational cost. Experiments across ViT-B/16 and ResNet-50 backbones show that Fly-CL achieves comparable or higher accuracy than prior representation-based methods (e.g., RanPAC, F-OAL) while reducing post-extraction training time.

**Strengths:**

1. The paper is clear motivated and formulated, easy-to-follow
2. High efficiency with strong accuracy trade-off
3. Simple and generalizable design, broadly applicable to backbones and datasets.

**Weaknesses:**

1. The proposed framework shares conceptual similarities with earlier representation-based approaches such as RanPAC and F-OAL, both of which employ random projections and analytic updates. While Fly-CL introduces additional sparsification and adaptive regularization, the methodological advancement over these predecessors appears incremental rather than fundamentally novel.

2. This paper does not empirically demonstrate the effect of reduced prototype correlation.
3. The experiments primarily report average accuracy across tasks, but omit standard CL metrics such as the final-task accuracy ($\mathbf{A}_T$) and forgetting measure.
4. The study focuses on representation-based and prompt-based baselines but do not compared with recent lora/adapter-efficient continual tuning methods such as InfLoRA [A], SEMA [B], and MoE-Adapters [C].
5. Discuss why Fly-CL slightly underperforms RanPAC on CIFAR-100 despite achieving substantial gains on other datasets.
6. It is not specified whether the ViT-B/16 model is initialized from ImageNet-21K or ImageNet-1K pre-trained weights.

[A] Liang, Y. S., & Li, W. J. Inflora: Interference-free low-rank adaptation for continual learning. CVPR2024.

[B] Wang, H., et al. Self-expansion of pre-trained models with mixture of adapters for continual learning. CVPR2025

[C] Yu, J., et al. Boosting continual learning of vision-language models via mixture-of-experts adapters. CVPR2024.

**Questions:**

see weakness.

---

> ### Author Response · Authors · 2025-11-27
> **Official Response to Reviewer uEtz from Author (Part I)**
>
> We appreciate the reviewer's insightful comments. In response to the valuable feedback, we have conducted extensive new simulations to thoroughly address the concerns.
>
> > The proposed framework shares conceptual similarities with earlier representation-based approaches such as RanPAC and F-OAL, both of which employ random projections and analytic updates. While Fly-CL introduces additional sparsification and adaptive regularization, the methodological advancement over these predecessors appears incremental rather than fundamentally novel.
>
> The reviewer is correct in noting the conceptual similarities between Fly-CL and earlier representation-based approaches such as RanPAC and F-OAL, although the structure of Fly-CL is inspired by the fly olfactory circuit. Our work offers several advantages over these prior methods:
>
> 1. **Computational Efficiency**
>
> While earlier methods achieve strong accuracy, their training inefficiency remains a major bottleneck for real-world deployment. Fly-CL addresses this challenge through sparsification and adaptive regularization, achieving comparable or superior accuracy with significantly improved efficiency.
>
> 2. **Biological Plausibility and Motivation**
>
> Unlike purely algorithmic predecessors, Fly-CL is biologically inspired, with components directly mapped to mechanisms in the Drosophila olfactory circuit (e.g., sparse coding, Hebbian-like learning). In **Appendix B.3**, we explicitly derive the connection between ridge classification and biologically plausible Hebbian rules, thereby bridging neuroscience and machine learning.
>
> 3. **Practical Impact**
>
> By improving efficiency while preserving accuracy, Fly-CL provides a deployable solution for resource-constrained scenarios, advancing the field from theoretical frameworks toward applicable, biologically inspired AI.
>
> We agree that innovation often emerges incrementally, but we believe that Fly-CL's combined contributions—efficiency gains, biological grounding, and scalability—represent a meaningful step forward. These points have been emphasized in the revised manuscript.
>
> > This paper does not empirically demonstrate the effect of reduced prototype correlation.
>
> We apologize that this critical aspect was not sufficiently emphasized in the original manuscript, and we have revised the text to highlight these findings more explicitly in **Section 4**.
>
> As demonstrated in **Figure 3**, the heatmaps of Pearson correlation coefficients between prototypes reveal a clear progression of decorrelation across the Fly-CL pipeline:
>
> 1. **Initial Feature Extraction** (Figure 3a): Prototypes exhibit strong correlations.
>
> 2. **After Random Projection & Top-k Sparsification** (Figure 3b): Correlations are visibly reduced.
>
> 3. **Following Ridge Classification** (Figure 3c): Decorrelation is further enhanced.
>
> These results provide **empirical validation** of the progressive decorrelation enabled by Fly-CL's architecture. We have revised the manuscript accordingly to ensure these insights are presented more prominently.
>
> > The experiments primarily report average accuracy across tasks, but omit standard CL metrics such as the final-task accuracy ($A_T$) and forgetting measure.
>
> To provide a more comprehensive evaluation of model performance, we have expanded our analysis beyond the overall accuracy (reported in Table 1) to include two additional key metrics in **Table 10**:
>
> - **Last Stage Accuracy ($A_T$)**: Measures the model's performance on the most recent task, reflecting its ability to acquire new knowledge.
>
> - **Backward Transfer (BWT)**: Quantifies the degree of forgetting by evaluating how learning new tasks affects performance on previous ones.
>
> > The study focuses on representation-based and prompt-based baselines but do not compared with recent lora/adapter-efficient continual tuning methods such as InfLoRA [A], SEMA [B], and MoE-Adapters [C].
>
> We appreciate the reviewer’s valuable feedback. In response to the concerns, we have conducted **comprehensive comparisons** with state-of-the-art **LoRA/adapter-based continual tuning approaches**, including **InfLoRA, SEMA, and MoE-Adapters**, as detailed in **Appendix D.2** of the revised manuscript.
>
> The experimental results (**Tables 1, 5, 6, 8, and 10**) demonstrate that **Fly-CL consistently outperforms these methods across multiple benchmarks**, achieving:
>
> - Higher accuracy,
>
> - Reduced training time, and
>
> - Lower memory overhead.
>
> These improvements highlight **Fly-CL's superior efficiency and scalability** for CL tasks. We have revised the manuscript accordingly to ensure these results are clearly presented.

---

> ### Author Response · Authors · 2025-11-27
> **Official Response to Reviewer uEtz from Author (Part II)**
>
> > Discuss why Fly-CL slightly underperforms RanPAC on CIFAR-100 despite achieving substantial gains on other datasets.
>
> Compared with Fly-CL, RanPAC incorporates a PEFT technique to adapt the pretrained model to the domain of the CL datasets prior to the continual learning stage. We hypothesize that, for CIFAR-100, this domain adaptation is particularly beneficial, as it allows the model to capture sample features more effectively during continual learning. This likely explains why RanPAC achieves slightly higher accuracy than Fly-CL on this dataset. However, the PEFT technique also introduces substantial memory overhead and increases training time, making it a double-edged sword.
>
> > It is not specified whether the ViT-B/16 model is initialized from ImageNet-21K or ImageNet-1K pre-trained weights.
>
> The ViT-B/16 checkpoint we used was pretrained on ImageNet-21K and subsequently fine-tuned on ImageNet-1K, a configuration commonly referred to in the literature as ViT-B/16-IN1K. We have clarified this in **Appendix E.1** of the revised manuscript, and the download link for this checkpoint is provided in **Table 12**.

---

### Official Review · Reviewer_gFjn · 2025-11-01

**Soundness:** 3
**Presentation:** 3
**Contribution:** 3
**Rating:** 6
**Confidence:** 4

**Summary:**

This paper proposes Fly-CL, a continuous learning framework inspired by the Drosophila olfactory circuit, designed to address multicollinearity issues in representation learning based on pre-trained models while reducing training time. Fly-CL achieves feature decoupling and efficient classification through mechanisms including sparse random projection, Top-k activation filtering, and streaming ridge classification. Experiments demonstrate that this method achieves or surpasses state-of-the-art performance across multiple datasets and backbone networks while significantly reducing training time, exhibiting particularly strong advantages during post-feature extraction processing stages.

**Strengths:**

1. The paper addresses the issue of “persistent feature decoupling in pre-trained models.” While the constituent components (random projection, Top-k, ridge regression) are established techniques, their creative combination and application constitute the contribution.

2. The paper provides a solid theoretical foundation, demonstrating the information retention capability of sparse projections. The experimental design encompasses multiple architectures (ViT, ResNet), datasets, and evaluation metrics.

3. This work provides a solution for efficient continuous learning in resource-constrained scenarios.

**Weaknesses:**

1. I believe the core mechanism of the paper—“random projection + Top-k sparse activation”—shares striking similarities with the fundamental concept of Kanerva's Sparse Distributed Memory (SDM). SDM is similarly inspired by neuroscience and employs high-dimensional sparse representations and similarity matching to address memory and learning challenges. The paper omits discussion with this classic approach.

2. Despite significant efficiency gains, projecting dimensions of m=10,000 may still impose memory constraints on extreme edge devices.

**Questions:**

1. Could the author establish a simple baseline by using only Fly-CL's projection and Top-k layers, followed by a straightforward linear classifier or k-nearest neighbors classifier, to demonstrate the necessity of streaming ridge regression in your problem setting?

2. Although Fly-CL claims to mitigate forgetting, how does it affect old tasks at the feature space level? When learning new tasks, do the class prototypes of old tasks drift or distort in the high-dimensional space after Fly-CL processing?

3. Theorem B.1 aims to prove that sparse projection matrices W are almost certainly full-rank, which is considered an argument for their ability to preserve information. However, in machine learning, the fact that a random matrix is full-rank does not directly equate to it being a “good” feature mapper. More importantly, the matrix's isometric property or distance-preserving characteristic is guaranteed by the Johnson-Lindenstrauss (JL) lemma. Can the authors provide evidence that your sparse matrix W indeed preserves pairwise distances between feature vectors with high probability?

---

> ### Author Response · Authors · 2025-11-27
> **Official Response to Reviewer gFjn from Author**
>
> We appreciate the reviewer's insightful comments. In response to the valuable feedback, we have conducted extensive new simulations to thoroughly address the concerns.
>
> > I believe the core mechanism of the paper—"random projection + Top-k sparse activation"—shares striking similarities with the fundamental concept of Kanerva's Sparse Distributed Memory (SDM). SDM is similarly inspired by neuroscience and employs high-dimensional sparse representations and similarity matching to address memory and learning challenges. The paper omits discussion with this classic approach.
>
> We appreciate the reviewer for directing us to the relevant line of work. As suggested, we have added a detailed discussion of SDM in **Appendix D.3**. Furthermore, we found that SDM can be implemented under our CIL setting. In our experiments, SDM’s parameters were updated using SGD with a learning rate of 0.01, and the overall accuracy $\bar{A}$ was reported under the same experimental settings as in Table 1. The results show that Fly-CL outperforms SDM by a substantial margin in accuracy.
>
> | Method | CIFAR-100  | CUB-200-2011 |    VTAB    |
> | :----: | :--------: | :----------: | :--------: |
> | Fly-CL | 93.89±0.12 |  93.84±0.18  | 96.54±0.38 |
> |  SDM   | 46.28±0.97 |  53.42±1.85  | 61.68±2.29 |
>
> > Despite significant efficiency gains, projecting dimensions of m=10,000 may still impose memory constraints on extreme edge devices.
>
> The reviewer is correct that resource constraints inevitably involve a trade-off with performance. However, as shown in Figure 5 and Table 9, reducing the projection dimension to m = 5,000 results in only a slight accuracy drop (typically around 1%), which is generally acceptable. Under this setting, Fly-CL incurs minimal additional memory overhead beyond that of the pretrained model. For scenarios with stricter memory limitations, a smaller pretrained backbone can be employed for feature extraction, making the approach suitable even for extreme edge devices.
>
> > Could the author establish a simple baseline by using only Fly-CL’s projection and Top-k layers, followed by a straightforward linear classifier or k-nearest neighbors classifier, to demonstrate the necessity of streaming ridge regression in your problem setting?
>
> We regret that our results were not presented clearly enough in the original manuscript. The baseline suggested by the reviewer corresponds to the **"w/o ridge"** variant reported in **Figure 4**. This baseline retains all components of Fly-CL except streaming ridge classification; instead, it performs classification solely via similarity matching with class prototypes. As shown in Figure 4, the substantial performance gap between "Fly-CL" and "w/o ridge" clearly demonstrates the necessity and effectiveness of streaming ridge regression in our method.
>
> > Although Fly-CL claims to mitigate forgetting, how does it affect old tasks at the feature space level? When learning new tasks, do the class prototypes of old tasks drift or distort in the high-dimensional space after Fly-CL processing?
>
> The reviewer is correct that the class prototypes of old tasks will be modified when learning new tasks, due to the additional correlations with the new class prototypes. The impact on old tasks at the feature-space level is equivalent to the impact on their corresponding class prototypes. As shown in Equation 6, all class prototypes (including those from old tasks) are recalculated. Fly-CL mitigates forgetting by reframing it as a similarity matching problem: by solving Equation 6, prototypes from both new and old tasks are jointly considered to produce a well-decoupled solution. This approach performs better than the vanilla method, which only updates prototypes for new tasks.
>
> > Theorem B.1 aims to prove that sparse projection matrices W are almost certainly full-rank, which is considered an argument for their ability to preserve information. However, in machine learning, the fact that a random matrix is full-rank does not directly equate to it being a "good" feature mapper. More importantly, the matrix's isometric property or distance-preserving characteristic is guaranteed by the Johnson-Lindenstrauss (JL) lemma. Can the authors provide evidence that your sparse matrix W indeed preserves pairwise distances between feature vectors with high probability?
>
> We have added additional simulation experiments in **Appendix B.1 (Figure 6)** of the revised manuscript to support the property of pairwise distance preservation. In Figure 6, we measure the distortion ratio between the normalized projection distance and the original distance between two points (ideally be 1). The results show that approximately 96.87% of pairwise distances fall within the range [0.97, 1.03], indicating that the distance-preserving property holds with high probability. Further experimental details can be found in Appendix B.1 of the manuscript.

---

### Official Review · Reviewer_4tXN · 2025-11-04

**Soundness:** 3
**Presentation:** 3
**Contribution:** 3
**Rating:** 6
**Confidence:** 3

**Summary:**

Taking inspiration from the fly's olfactory system, the authors introduce FLY-CL, a way to extract and use features of a pretrained model for class incremental learning.

The features of a pretrained model are expanded with a fixed random projection, in a similar way to the a layer from "projection neurons" to "Keynyon cells" in the olfactory circuit of a fly. A top-k activation simulates lateral inhibition. A learned similarity matching down projection emulates the projection to "mushroom body output neurons".

They show how this method reduces catastrophic forgetting when using pretrained models with a set of image benchmarks.

**Strengths:**

The high-dimensional sparse layer does seem to effectively prevent catastrophic forgetting.

No task identity is required at inference.

works with unmodified pre-trained models

ablation studies do show the need for each part of the model (random projection / ridge regression / normalisation)

Figure 5 shows how important the high dimension layer is, with performance saturating at m>10k. This is crucial since I believe this is the main novel contribution.

**Weaknesses:**

There can be high memory costs associated with the large sparse layer (the authors do discuss this).

They do need the model to receive task boundaries and to store "class prototypes" for use during inference.

No adaptation to the backbone so it is dependent on a good pretrained model (a weakness or a strength depending on the circumstance).

The paper would benefit from a more detailed analysis of how the key hyperparameters (m, p, and k) scale with task complexity, the number of tasks/classes, and different pretrained backbones.

minor typo:
037 "generalization in downstream tasks for downstream tasks"

**Questions:**

you show the performance saturates beyond 10k. Do you expect that saturation point to remain stable as the number of tasks/classes grows much larger?

Have you run experiments that study how the required projection dimensionality scales with increased task complexity or number of tasks/classes? Or if it changes if you use a different pretrained backbone?

---

> ### Author Response · Authors · 2025-11-27
> **Official Response to Reviewer 4tXN from Author**
>
> We appreciate the reviewer's insightful comments. In response to the valuable feedback, we have conducted extensive new simulations to thoroughly address the concerns.
>
> > There can be high memory costs associated with the large sparse layer (the authors do discuss this).
>
> The reviewer is correct on this point; however, our empirical results show that performance saturates once the expansion dimension exceeds 10,000 (Figure 5), well before memory consumption increases substantially (Table 9). Therefore, adopting an excessively high dimension is unnecessary. A moderate expansion dimension m provides an effective balance between performance and efficiency in Fly-CL.
>
> Moreover, under this setting, Fly-CL already consumes significantly less memory than previous methods (Table 8), indicating that this concern has been largely addressed.
>
> > They do need the model to receive task boundaries and to store "class prototypes" for use during inference.
>
> We apologize for the misunderstanding. Fly-CL does not require task boundaries during inference. In the CIL setting, all methods must predict the class ID of a test sample without prior knowledge of the task to which the class belongs, and Fly-CL adheres to this protocol. We have clarified this point in Section 3.1 of the revised manuscript.
>
> Although Fly-CL stores class prototypes, this mechanism operates analogously to the classification head in other methods during inference and does not introduce any notable increase in inference time or memory usage compared to a single classification head in alternative approaches.
>
> > No adaptation to the backbone so it is dependent on a good pretrained model (a weakness or a strength depending on the circumstance).
>
> We agree that the no-adaptation design makes it important to select an appropriate pretrained model for practical deployment. Encouragingly, a wide range of pretrained models is readily available online, making it feasible to choose one well-suited for downstream tasks.
>
> > The paper would benefit from a more detailed analysis of how the key hyperparameters (m, p, and k) scale with task complexity, the number of tasks/classes, and different pretrained backbones.
>
> In response to this suggestion, we conducted additional sensitivity analyses of structural hyperparameters under three complementary settings: (i) ImageNet-A with 10 tasks and 20 classes per task using ViT-B/16, to examine the effect of task complexity (**Figure 10**); (ii) CUB-200-2011 with 20 tasks and 10 classes per task using ViT-B/16, to evaluate the impact of a larger number of tasks/classes (**Figure 11**); and (iii) CUB-200-2011 with 10 tasks and 20 classes per task using ResNet-50, to assess the influence of different backbone architectures (**Figure 12**). Across all settings, the observed trends remain consistent with those reported in Figure 5. The complete results are provided in **Appendix C.7**.
>
> > minor typo: 037 "generalization in downstream tasks for downstream tasks"
>
> Thanks for pointing out, we have fixed it in the revised version.
>
> > you show the performance saturates beyond 10k. Do you expect that saturation point to remain stable as the number of tasks/classes grows much larger?
>
> The answer is yes, according to our additional results in **Appendix C.7 (Figure 11)**. The saturation point consistently occurs around 10,000 and does not noticeably increase as the number of tasks or classes grows.
>
> > Have you run experiments that study how the required projection dimensionality scales with increased task complexity or number of tasks/classes? Or if it changes if you use a different pretrained backbone?
>
> According to our additional results in **Appendix C.7 (Figures 10 and 12)**, the required projection dimensionality does not change significantly with increased task complexity, a larger number of tasks or classes, or when using ResNet-50.

---

### Author Response · Authors · 2025-11-27
**Sincerely Looking Forward to the Reviewers' Feedback**

Dear Reviewers,

We sincerely appreciate the depth and thoughtfulness of your comments, and we greatly value the time and effort you have dedicated to evaluating our work.

We have provided detailed responses to all reviewer questions and have updated the manuscript accordingly, with **all revisions highlighted in blue** throughout the paper.

If you have any further questions or require additional clarification, please feel free to let us know. We are grateful for the opportunity to continue the discussion if needed.

Best regards,

The Authors

---

### Meta-Review · Area_Chair_XrKB · 2026-01-02

**Summary:**

The reviewers broadly agreed that Fly-CL presents a well-engineered and efficient continual learning framework that achieves substantial training speedups while maintaining competitive accuracy. Initial concerns focused on the degree of methodological novelty relative to prior representation-based methods, the clarity of the biological motivation, the robustness of hyperparameter choices, and the completeness of the experimental evaluation (including additional metrics and baselines). The rebuttal substantially strengthened the paper through new experiments, empirical validation, further comparisons to SOTA methods, and improved theoretical and biological grounding. As a result, the AC recommends **acceptance** of the paper, congratulations.

**Reviewer Concerns:**

Concerns that were comprehensively addressed:

- Hyperparameter sensitivity and scalability (Reviewers **4tXN**, **awjr**): The authors added sensitivity analyses across the number of tasks/classes, different backbones, and larger foundation models, showing stable trends and saturation behavior.
- Relation to prior work (SDM, RanPAC, F-OAL) (Reviewers **gFjn**, **uEtz**): The rebuttal added explicit discussion, new baselines, and empirical comparisons showing both conceptual distinctions and clear performance gaps.
- Distance preservation and theoretical justification (Reviewer **gFjn**): Additional simulations provided evidence supporting approximate distance preservation of the sparse projection.
- Missing metrics and baselines (Reviewer **uEtz**): Continual learning metrics (last-task accuracy, backward transfer) and comparisons to InfLoRA, SEMA, and MoE-Adapters were added and strengthened the evaluation.
- Biological motivation and plausibility (Reviewer **awjr**): The revised manuscript strengthens the biological motivations.

Concerns that partially remain but are no longer blocking:
- Incremental novelty (Reviewer **uEtz**): While the method builds on existing components, the combination yields a practically meaningful advance in efficiency and robustness. This is now clearly supported empirically.
- Memory footprint for extreme edge settings (Reviewers **4tXN**, **gFjn**): The trade-off remains inherent, but the authors provide rational and reasonable operating points with minimal accuracy loss.

**Reviewer Scores:**

Reviewer **uEtz** is likely to increase from $4 \to 6$, given that major concerns were directly addressed. Reviewers **4tXN**, **gFjn**, and **awjr** are expected to maintain or increase their score ($\approx 6 \to 8$), given the authors' responses.

---

### Decision · Program_Chairs · 2026-01-26

Accept (Poster)